# LEARNING ON THE JOB: TEST-TIME CURRICULA FOR TARGETED REINFORCEMENT LEARNING

## ABSTRACT

Humans are good at learning on the job: We learn how to solve the tasks we face as we go along. Can a model do the same? We propose an agent that assembles a task-specific curriculum, called *test-time curriculum* (TTC-RL), and applies reinforcement learning to continue training the model for its target task. The test-time curriculum avoids time-consuming human curation of datasets by automatically selecting the most task-relevant data from a large pool of available training data. Our experiments demonstrate that reinforcement learning on a test-time curriculum consistently improves the model on its target tasks, across a variety of evaluations and models. Notably, on challenging math and coding benchmarks, TTC-RL improves the pass@1 of `Qwen3-8B` by approximately 1.8x on AIME25 and 2.1x on CodeElo. Moreover, we find that TTC-RL significantly raises the performance ceiling compared to the initial model, increasing pass@8 on AIME25 from 40% to 62% and on CodeElo from 28% to 43%. Our findings show the potential of test-time curricula in extending the test-time scaling paradigm to continual *training* on thousands of task-relevant experiences during test-time.

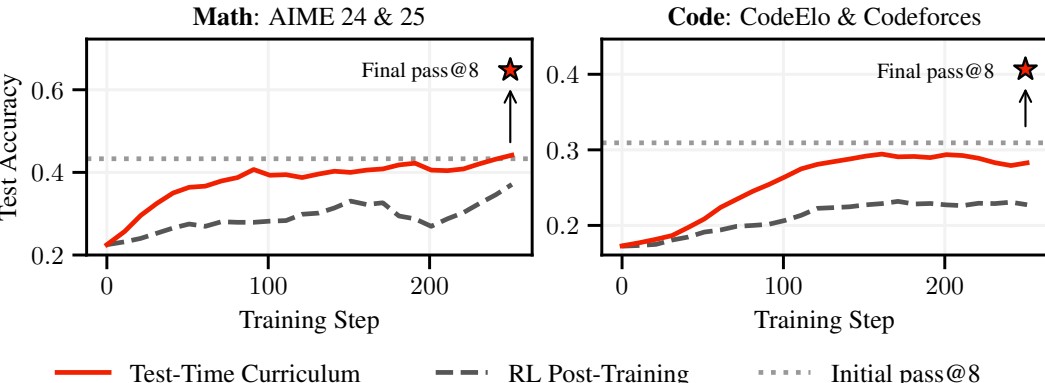

Figure 1: **Test-time curricula (TTCs) lead to remarkable improvements in math and coding by practicing on *self-curated* task-related problems at test-time.** The plots show the pass@1 test accuracy of `Qwen3-8B` throughout its test-time training. Our method, TTC-RL (solid red line), consistently improves performance, learning faster and achieving a higher final accuracy than standard RL post-training (dashed gray line). Notably, the final pass@1 accuracy of TTC-RL approaches the model's initial pass@8 performance (dotted gray line), which represents a proxy for the performance ceiling of the initial model. The stars indicate the final pass@8 values after TTC-RL, demonstrating a significant improvement over the initial pass@8, which indicates that the model learns new solution strategies at test-time.

## 1 INTRODUCTION

We study how large language models (LLMs) can continually improve at reasoning on their target tasks at test-time. Increasing test-time compute, for example, by extended use of context as scratch space, has recently emerged as a key direction for improving LLMs on challenging tasks such as math and coding (Jaech et al., 2024; Guo et al., 2025; Kimi et al., 2025). Test-time scaling has been

driven primarily by extensive general-purpose reinforcement learning (RL; Guo et al., 2025), where the LLM learns how to effectively use its context for reasoning. However, since the context of LLMs is bounded and becomes exceedingly expensive to expand, an LLM cannot learn in-context from experience over long timeframes.

One promising technique for overcoming this challenge is test-time training (TTT; Sun et al., 2020; Hardt & Sun, 2024), which continues training the model at test-time after being given a task. Previous work has studied TTT via supervised fine-tuning on human-created or expert data, either retrieved (Hardt & Sun, 2024; Hübotter et al., 2025) or provided as few-shot examples (Akyürek et al., 2025). Other work has instead focused on TTT in the context of recurrent neural networks (Sun et al., 2025; von Oswald et al., 2025; Zhang et al., 2025b), aiming to replace the costly attention-based context in Transformers (Vaswani et al., 2017) with a fixed-size state (i.e., the model itself), but losing some of the advantages of reasoning over an uncompressed scratchpad. We explore a complementary approach to test-time scaling, where an LLM is continually *trained* on self-curated training tasks related to its target task, while practicing on each individual training task in-context. This leverages the Transformer's attention as an uncompressed scratchpad for short-term ideation, while meta-learning strategies for leveraging that context across long-term, task-specific experience.

We propose a *test-time curriculum* (TTC) agent that automatically designs its own curriculum of training tasks by selecting the relevant tasks for the job from a large corpus of existing tasks. The agent then attempts tasks in its curriculum, and compresses the gathered experience into its weights via RL. The automatic self-guided curriculum design avoids laborious human curation of datasets, and enables training on purpose-built curricula at test-time. We find that this *reinforcement learning on test-time curricula* (TTC-RL) leads to remarkably improved reasoning on target tasks. In particular, we find that TTC-RL improves the pass@1 of several strong LLMs across diverse reasoning tasks, covering competition math, coding, and scientific reasoning (cf. Figure 1). We further identify that TTC-RL is complementary to other means of test-time scaling, effectively improving pass@$k$ and maj@$k$ even at large $k$. Notably, we find that TTC-RL can overcome the limitation of fixed context windows by observing that a non-thinking model (limited to 8k context tokens) with TTC-RL can perform similarly to the same model thinking for 30k tokens in-context. This demonstrates that during TTC-RL, the model continues *learning* how to think effectively for its target tasks. Our results suggest such targeted RL as a promising new direction for LLM agents that continually improve at test-time through many interactions with an environment.

We summarize our contributions as follows:

1. **We propose a TTC agent for targeted RL (§3):** We propose a test-time curriculum agent which at test-time when given a target task, self-selects related training tasks from a diverse corpus. The agent then learns from its own experience of attempting those tasks via RL.

2. **TTC-RL improves reasoning on target tasks (§4):** Across several models and tasks, TTC-RL consistently improves pass@1 substantially faster than general-purpose RL post-training on standard RL datasets, and saturates at a higher accuracy. Next, we identify that TTC-RL substantially raises the performance ceiling of the model (pass@$k$) and demonstrate that it is complementary to existing approaches to test-time scaling. Finally, we find that TTC-RL yields strongly specialized models that perform remarkably well on their target tasks, even when compared to models that are allowed to think for tens of thousands of tokens in context.

3. **Measuring latent improvements in reasoning (§5):** The evaluation of RL-trained models faces the challenge of estimating whether improved scores are due to better reasoning or merely learning the expected output format. We introduce a new metric, *latent improvement*, which computes a lower bound on the improvement in reasoning due to RL training, and find that TTC-RL leads to substantial improvements in "latent" reasoning.

## 2 RELATED WORK

**Test-time scaling and general-purpose RL training.** A common strategy for improving LLM performance in challenging domains is to allocate additional test-time compute, for instance, through majority voting (Snell et al., 2025), search with a reward model (Lightman et al., 2023; Wang et al., 2024a; Setlur et al., 2025a), or by identifying consistent patterns among parallel rollouts (Wang et al., 2023; Huang et al., 2025a). The potential of such methods is often measured by pass@$k$, which describes the performance ceiling with $k$ generations (Chen et al., 2025b).

More recently, scaling test-time compute via in-context "reasoning" (Brown et al., 2020; Wei et al., 2022) has significantly improved performance in domains like math and coding (Jaech et al., 2024). This capability is commonly enabled by large-scale, general-purpose RL training on diverse tasks (Lambert et al., 2025; Ma et al., 2025; Guo et al., 2025; Kimi et al., 2025), during which models learn to reason within their bounded context (Setlur et al., 2025b), which connects to the broad topic of meta-learning (Schmidhuber, 1987; Duan et al., 2017; Finn et al., 2017). Curriculum learning, first proposed by Bengio et al. (2009), has been successfully applied to challenging RL problems (Sinapov et al., 2015; Narvekar et al., 2020). This paradigm is closely related to goal-conditioned RL (Schaul et al., 2015; Andrychowicz et al., 2017) where several works have studied automatic curriculum generation (Florensa et al., 2018; Warde-Farley et al., 2018; Pitis et al., 2020; Pong et al., 2020). In contrast to improving general-purpose models, our work employs RL to train specialized reasoners for a particular target task at test-time.

**Self-play.** A specialized form of curriculum learning has proven highly successful in domains like games through the use of self-play (Schmidhuber, 1991; Silver et al., 2016), where an agent is repeatedly challenged by playing against itself. Seminal works show that this approach can lead to superhuman performance (e.g., Mnih et al., 2015; Silver et al., 2016; 2017; Berner et al., 2019). Several recent works aim to generalize this paradigm to LLMs and more general domains such as coding by self-generating a training curriculum (Zhao et al., 2025; Huang et al., 2025b; Chen et al., 2025a; Fang et al., 2025). While recent work has studied test-time curricula as an extension of self-play to goal-conditioned RL settings (Diaz-Bone et al., 2025), its evaluation has focused on simple robotic navigation tasks. We extend this line of work to challenging reasoning tasks by self-curating a training curriculum, enabling LLMs to continually learn from extensive experience on a single task (Silver & Sutton, 2025; Shen et al., 2025).

**Test-time training and test-time RL.** Training a model at test-time for a given input has been widely studied as TTT (Sun et al., 2020), using supervised (Hardt & Sun, 2024; Hübotter et al., 2025; Yu et al., 2025a; Bertolissi et al., 2025; Bagatella et al., 2025a) or self-supervised losses (Sun et al., 2025; Dalal et al., 2025). Several methods perform TTT in a purely unsupervised manner, i.e., without "real-world" data or feedback (Wang et al., 2021; Zhang et al., 2022). Most relevant to our work, Zuo et al. (2025) recently extended unsupervised TTT to perform RL on the test set, leveraging the model's majority votes as pseudo-labels. This connects to a broader theme of unsupervised RL (Zhang et al., 2025a; Shao et al., 2025; Zhou et al., 2025; Prabhudesai et al., 2025) and self-improvement in LLMs (Zelikman et al., 2022; Gulcehre et al., 2023; Lee et al., 2025).

## 3 TEST-TIME CURRICULA

We consider the set of *target tasks* $\mathcal{D}^\star = \{x_1^\star, \ldots, x_M^\star\}$ given at test-time, and our goal is to specialize an existing model through further training to those tasks. For training, as in general-purpose RL, we rely on an existing large corpus of training tasks $\mathcal{D} = \{(x_i, v_i)\}_{i=1}^N$, for each of which $v_i(\cdot) \in \{0, 1\}$ verifies whether an attempt was correct. To specialize, it is common practice to construct a particular subset $\widehat{\mathcal{D}}^\star$ from $\mathcal{D}$, and we call such a targeted subset a *test-time curriculum* for $\mathcal{D}^\star$. We seek to make test-time training on such a curriculum scalable. To this end, we propose to go beyond human-curated test-time curricula and let the initial model craft its own test-time curriculum.

We propose a test-time curriculum agent, outlined in Algorithm 1. In each training step, the agent selects its next training task from the corpus $\mathcal{D}$ based on its target tasks $\mathcal{D}^\star$ and the current model $\theta_{t-1}$. This step leverages the semantic understanding of the model to *self-curate* a test-time curriculum for the target tasks. We then train on this test-time curriculum via GRPO (Shao et al., 2024).[1] Note that test-time training does not necessitate the model to stay

---

**Algorithm 1** Test-Time Curriculum for Targeted RL

---
**Require:** Test tasks $\mathcal{D}^\star$
1: **for** $t = 1, 2, \ldots, T$ **do**
2:      $(x_t, v_t) \leftarrow \text{TTC}_{\theta_{t-1}, \mathcal{D}}(\mathcal{D}^\star)$     ▷ select next task
3:      $\{\hat{y}_{t,i}\} \sim \pi_{t-1}(\cdot \mid x_t)$           ▷ attempt
4:      $\{r_{t,i}\} \leftarrow v_t(\{\hat{y}_{t,i}\})$            ▷ verify
5:      $\theta_t \leftarrow \text{GRPO}(\theta_{t-1}, \{\hat{y}_{t,i}\}, \{r_{t,i}\})$    ▷ RL step
6: **end for**

---

[1] Algorithm 1 abstracts that we perform each RL step over a batch of training tasks and that we perform RL training for multiple episodes.

close to its initialization since it needs to generalize only to its target tasks, and hence, we omit the KL penalty of GRPO. We include background on GRPO in Appendix B.

While the previous works of Hardt & Sun (2024) and Hübotter et al. (2025) have studied self-curated test-time curricula with supervised fine-tuning (SFT) and shown that this can improve language modeling (i.e., lead to lower perplexity), we find that this approach does *not* improve accuracies on reasoning tasks. Perhaps counterintuitively, we find that test-time training with SFT—even on correct demonstrations of test tasks—can lead to an initial performance drop. Our findings mirror recent observations on the robustness of on-policy RL compared to off-policy SFT (Shenfeld et al., 2025). We provide further details in Appendix A. Moreover, while test-time training via SFT requires the corpus to specify *how* training tasks are to be solved, test-time training via RL only requires *verification* of solutions.

We next describe the corpus and TTC method used in our evaluation of the TTC-RL setting.

**An automatic TTC for targeted RL.** We adopt existing methods for TTC selection from previous work studying test-time curricula with SFT (Hardt & Sun, 2024; Hübotter et al., 2025). These methods leverage a latent representation space $\phi$ over token sequences for which we use the normalized last-token last-layer embeddings of the initial model. We then utilize SIFT (Hübotter et al., 2025) which selects those examples from the corpus that the model deems most informative for the target tasks. SIFT has a hyperparameter $\lambda$ which explicitly trades between diversity of the selected examples and their relevance to the target tasks. We find that our results are robust to the choice of $\lambda$ and generally set $\lambda = 0.1$ in our experiments. Appendix F gives examples for such self-curated test-time curricula and we include background on SIFT in Appendix B.

The TTC selected by SIFT is static for given target tasks. Motivated by previous work on curricula for RL (e.g., Florensa et al., 2018; Narvekar et al., 2020; Pitis et al., 2020; Zhao et al., 2025) we also evaluate an adaptive curriculum that selects training tasks of appropriate difficulty for the current model. We find that this leads to diminishing returns if the corpus difficulty is already appropriately calibrated to the initial model, and therefore focus on the static curriculum in our main experiments. In Appendix C, we demonstrate that using a TTC of appropriately challenging training tasks, can significantly accelerate learning for a weaker initial model such as `Qwen3-0.6B`.

**A diverse corpus for general-purpose RL post-training.** To study the effectiveness of our proposed adaptive test-time curriculum, we leverage a large corpus of high-quality verifiable training data, suitable for post-training a model across diverse domains. We assemble a new meta-dataset, which we call the `verifiable-corpus` and which combines approximately 265k diverse training tasks, spanning three environments:

- **Exact answer match / Math:** For math problems with a numerical answer, we determine answer equivalence using `math-verify`. Our corpus contains the training splits of GSM8K (Cobbe et al., 2021) and MATH (Hendrycks et al., 2021b), and the DAPO math dataset (Yu et al., 2025b), covering numerically verifiable math problems for a wide range of difficulties.

- **Judged answer match / General reasoning:** Measuring the validity of complex reasoning requires more robust verification than symbolic equivalence checks. Given a (potentially long) golden answer, we use a 1.5B-parameter verifier model trained by Ma et al. (2025) to determine whether attempted and golden answers are semantically equivalent. Our corpus contains the Webinstruct-verified dataset (Ma et al., 2025), which covers a wide variety of subjects ranging from natural sciences to history.

- **Unit tests / Code:** Finally, we combine several sources of coding tasks. Each coding task is verified by a set of unit tests. Our corpus combines tasks from APPS (Hendrycks et al., 2021a), code contests (Li et al., 2022), TACO (Li et al., 2023), PrimeIntellect (Mattern et al., 2025), Leetcode (Xia et al., 2025), the Codeforces training split (Penedo et al., 2025) and all LiveCodeBench tasks (Jain et al., 2025) prior to February 1, 2025.

We perform a filtering step where we remove training tasks with empty answers or less than 5 unit tests, to ensure a reliable training signal. Finally, we deduplicate and decontaminate the corpus, as detailed in Appendix E.1. We openly share the corpus and our environment implementations to support future research. To our knowledge, the `verifiable-corpus` is one of the first public corpora of high-quality verifiable tasks, spanning several domains and environments. We envision that, building on this work, future efforts will ultimately enable TTC agents to utilize any relevant

| Model | AIME24 | AIME25 | MATH500 | Codeforces | CodeElo | LCB$^{\text{v6}}$ | GPQA-D |
|---|---|---|---|---|---|---|---|
| **Qwen3-8B** | 21.67 | 23.33 | 69.55 | 20.85 | 13.73 | 20.61 | 49.11 |
| + RL post-training | 41.67 | 38.33 | 82.50 | 27.83 | 22.67 | 25.95 | 56.47 |
| + TTC-RL | **50.83**$^{+29.2}$ | **41.67**$^{+18.3}$ | **85.10**$^{+15.6}$ | **33.35**$^{+12.5}$ | **29.34**$^{+15.6}$ | **27.29**$^{+6.7}$ | **58.38**$^{+9.3}$ |
| **Qwen3-4B-Instruct-2507** | 52.50 | 40.83 | 72.00 | 26.70 | 20.27 | 21.56 | 61.93 |
| + RL post-training | 55.83 | **47.50** | 86.30 | 28.39 | 21.18 | 25.95 | **62.82** |
| + TTC-RL | **60.00**$^{+7.5}$ | 45.83$^{+5.0}$ | **88.50**$^{+16.5}$ | **34.99**$^{+8.3}$ | **27.20**$^{+6.9}$ | **26.91**$^{+5.4}$ | 61.93$^{+0.0}$ |
| **Qwen3-8B-Base** | 15.83 | 14.17 | 63.10 | 9.92 | 6.67 | 11.26 | 29.70 |
| + RL post-training | 22.50 | 20.83 | 76.85 | 17.46 | 9.97 | **18.51** | 42.77 |
| + TTC-RL | **30.00**$^{+14.2}$ | **21.67**$^{+7.5}$ | **78.15**$^{+15.1}$ | **17.84**$^{+7.9}$ | **11.33**$^{+4.7}$ | 17.94$^{+6.7}$ | **45.94**$^{+16.2}$ |

Table 1: **Performance of TTC-RL on reasoning benchmarks.** We evaluate TTC-RL across benchmarks for math (AIME24, AIME25, MATH500), coding (Codeforces, CodeElo, LCB$^{\text{v6}}$), and scientific reasoning (GPQA-D). Numbers in **bold** denote the best performance for a given model backbone, and we use + to denote the improvement over the initial model in percentage points.

training tasks they find on the web (similarly to retrieval-augmented generation; Lewis et al., 2019), or to self-generate their own training tasks (see, e.g., Zhao et al., 2025).

# 4 RESULTS

We focus our evaluation on a diverse set of target tasks in math, coding, and scientific reasoning. Specifically, we evaluate test-time curricula for high-school-level competition math questions in AIME 24 & 25 and MATH500 (Hendrycks et al., 2021b). We evaluate coding ability on Codeforces (Penedo et al., 2025), CodeElo (Quan et al., 2025), and on LiveCodeBench v6 (Jain et al., 2025), i.e., tasks released after February 1, 2025. Finally, we evaluate scientific reasoning with GPQA-Diamond (Rein et al., 2024) which covers questions in biology, physics, and chemistry.

TTC-RL can be applied to each task within a benchmark individually or to the entire benchmark on aggregate, treating it as a set of target tasks. We primarily evaluate TTC-RL per-benchmark as this yields greater statistical significance under a limited compute budget. We then perform an ablation, indicating that per-task TTCs performs at least on-par with per-benchmark TTCs (cf. Section 4.2).

To ensure that our evaluation is accurate, we adopt evalchemy (Raoof et al., 2025) and synthesize system prompts to be consistent across benchmarks (cf. Appendix E.2). We generally train for two episodes with batch size 8 and 16 rollouts per train task,[2] and measure avg@4 on the set of test tasks once every ten steps. To further reduce noise, we compute a moving average across three validation steps. Finally, in our summarized numeric results, we report the highest averaged avg@4, and include detailed plots of avg@4 per step in Appendix D.2.

We perform our main evaluation on the non-thinking models Qwen3-8B (Yang et al., 2025) and the more recent Qwen3-4B-Instruct-2507, whose responses we limit to 8192 tokens. We additionally evaluate on the Qwen3-8B base model. We opt for non-thinking models due to the high computational cost of running thinking models over long contexts, typically of up to 32k tokens. The goal of our TTC framework is to show that models can improve at test-time, even without further expanding their context. We hypothesize that our results extend to thinking models, which simply have a larger maximum response length.

**Main results.** We summarize our main results in Figure 1 and Table 1. We find that TTC-RL significantly improves accuracy across a range of models and all benchmarks. Notably, it also leads to significant performance gains on top of Qwen3-8B-Base within only relatively few RL steps, indicating that TTCs lead to sample-efficient training. Our main baseline is a model that is trained on 1k uniformly chosen training tasks from the corpus, to which we refer to as standard "RL post-training", since this method yields a general-purpose model. We compare this to TTC-RL with a curriculum of size 1k and find that training on a test-time curriculum accelerates learning significantly and leads to

---

[2]We summarize all training hyperparameters in Appendix E.3.

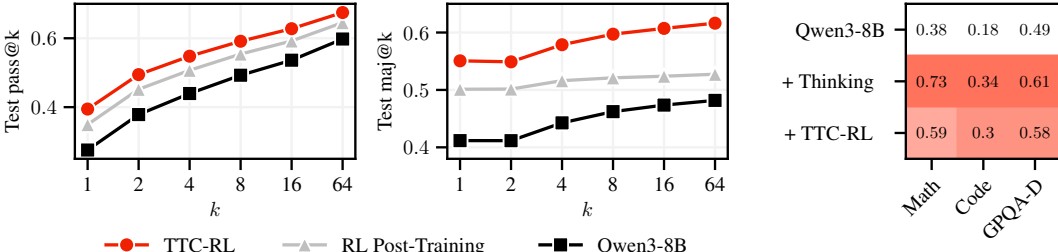

Figure 3: **TTC-RL scales test-time compute in way that is complementary to other means of test-time scaling. Left:** The pass@$k$ of TTC-RL on Qwen3-8B, averaged over benchmarks, increases substantially for small and large $k$, indicating that TTC-RL raises the model's performance ceiling. **Middle:** TTC-RL also improves the performance of majority voting (across math and GPQA-D), with the initial pass@1 significantly outperforming maj@64 on the initial model. **Right:** We evaluate Qwen3-8B in non-thinking and thinking mode, as well as the non-thinking model + TTC-RL. The color indicates the relative accuracy per column. We find that TTC-RL significantly improves the non-thinking model, allowing it to perform close to the thinking variant in several domains, despite reasoning over 8k rather than 30k context tokens.

saturation at substantially higher performance.[3] Notably, Qwen3-8B with TTC-RL performs on-par with strong closed-source non-thinking models; for example, it approximately matches GPT-4o-2024-08-06 on LCB[v6] and outperforms GPT 4.1 and Claude Opus 4.1 on AIME.

In Figure 2, we further ablate the size of the curriculum and find that TTC-RL consistently outperforms general-purpose RL post-training across a wide range of curriculum sizes. Interestingly, at dataset size 1—though performing poorly—the general-purpose RL post-training outperforms TTC-RL. We suspect that this may result from TTC-RL picking a practice task that is very similar to the test tasks, in which case overfitting may harm more than when overfitting to a less related task.

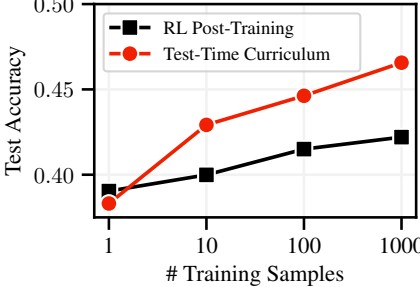

Figure 2: TTC-RL outperforms RL post-training across data sizes. We evaluate Qwen3-8B on all seven benchmarks and report the average test accuracy when training for 250 steps.

> **Takeaway 1**
>
> TTC-RL substantially improves accuracy on a wide variety of models and benchmarks, compared to a model's initial performance and after (continued) RL post-training on our corpus.

### 4.1 TTCs ARE COMPLEMENTARY TO EXISTING APPROACHES TO TEST-TIME SCALING

Next, we demonstrate that TTC-RL *improves* the LLM's ability for test-time scaling.

**TTCs raise the model's performance ceiling.** While the improvement in accuracy demonstrates that during TTC-RL, the model learns to better reason within context, we ask whether the model improves more broadly. A common metric to understand a model's "performance ceiling" for test-time scaling is the pass@$k$ metric, which measures whether any one of $k$ attempts is correct (Chen et al., 2025b). Recent work has repeatedly shown that RL-training tends not to improve pass@$k$ at large $k$ (Yue et al., 2025), leading to the concern that RL-training is simply "distilling" pass@$k$ into pass@1. In Figure 3 (left), we instead observe that TTC-RL significantly improves pass@$k$ across a wide range of $k$. Similarly, TTC-RL also improves the realized performance gains of majority voting, as can be seen in Figure 3 (middle), and notably increases the pass@1 well beyond the maj@64 after continued RL post-training. Since RL post-training and TTC-RL differ only in their training tasks, our results demonstrate that targeted selection of training tasks can lead to

---

[3]In Appendix D.3, we additionally compare to an "RL post-training" baseline that only samples training tasks from the test environment and show that this yields comparable results.

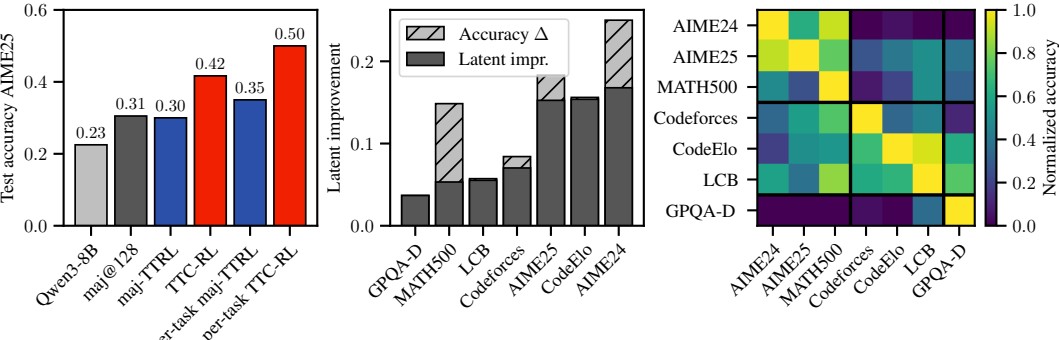

Figure 4: **Left: Per-task TTC-RL outperforms a benchmark-level TTC in AIME25.** We perform TTC-RL and maj-TTRL (cf. Section 5.2) on `Qwen3-8B`, and find that per-task TTC-RL even outperforms the benchmark-level TTC. **Middle: TTC-RL improves "correctness" of reasoning, not only learning the answer format.** We evaluate the difference in accuracy between TTC-RL and the initial `Qwen3-8B`, averaged over benchmarks. The latent improvement is a lower bound on the accuracy gain that is not due to merely learning the format (cf. Section 5.1). **Right: TTC-RL yields models that are specialized to their target tasks.** We plot the accuracy of `Qwen3-8B` trained for given target tasks (rows) when evaluated on other benchmarks (columns). We normalize accuracies across all evaluations of a particular benchmark. Notably, the model trained via TTC-RL for the "right" target tasks (i.e., the diagonal) always performs best.

substantial performance gains. Furthermore, we find that increasing clip-high in GRPO as proposed by Yu et al. (2025b) improves exploration and prevents entropy collapse in RL post-training as well as TTC-RL (cf. Appendix D.1), which is crucial for improving upon the strong initial models. Developing a better understanding of the circumstances under which RL-training can "discover new behavior", leading to improved pass@$k$, is an exciting direction for future research.

**TTC-RL with a short-context LLM can perform close to a long-context LLM.** We also seek to better understand how TTC-RL relates to reasoning over long contexts. To this end, we evaluate the non-thinking and thinking variants of `Qwen3-8B`, limited to 8k and 30k tokens per response, respectively. In Figure 3 (right), we find that TTC-RL on the non-thinking model performs close to the thinking model in several domains, particularly in coding and GPQA.[4] Further, note that the asymptotic cost of growing context in a Transformer is quadratic (Vaswani et al., 2017), whereas the asymptotic cost of TTC-RL is linear (since experience is compressed into the model's weights). This suggests that there is a regime in which, given a fixed compute budget, TTC-RL outperforms further scaling of context size. We believe that studying this compute-optimal Pareto frontier is an exciting topic for future research. Our results indicate that to further improve the performance of LLMs, test-time curricula may eventually be advantageous over continued scaling of context size.

> **Takeaway 2**
>
> Test-time curricula substantially increase the pass@$k$ performance ceiling of a model and can perform similarly to models which are reasoning over a much larger context. This indicates the potential of TTCs to complement existing approaches to test-time scaling.

### 4.2 TTCs EFFECTIVELY SPECIALIZE MODELS

To determine whether the test-time curriculum specializes the model to its target tasks, we conduct a straightforward experiment: We evaluate each final checkpoint of TTC-RL on all benchmarks, including those that were not part of the set of target tasks. We summarize the results in Figure 4 (right), with columns corresponding to evaluation and rows corresponding to training. We find that after TTC-RL, models perform best on their target tasks, while severely underperforming on tasks that are unrelated to the target tasks. Moreover, we identify a block-diagonal structure, where models generalize better across mutually related groups of tasks, particularly among similar math

---

[4]In MATH500, non-thinking `Qwen3-8B` + TTC-RL (85%) even outperformed the thinking variant (77%).

benchmarks. We also find that models appear to generalize better from coding to math than vice versa, and models generalize better from code and math to GPQA than vice versa.

**TTCs for individual tasks.** Aspirationally, we anticipate test-time curricula to enable continual learning for a single test task over a long timeframe. While we focus our main evaluation on the setting where test-time curricula are applied per benchmark, we run an ablation with 30 separate TTCs—one per AIME 25 question. The results in Figure 4 (left) demonstrate that *specializing* to an individual test task can outperform a broader specialization to a group of test tasks. This shows that TTC-RL does not depend on a larger set of test tasks to implicitly lead to diverse data and robust training, and instead seamlessly extends to a fully test-time setting with only a single task given. We find, however, that more fine-grained specialization does not always lead to further performance gains. We evaluate training separate TTCs for each of biology, physics, and chemistry in GPQA, leading to approximately the same performance as a joint TTC. In our view, gaining a better understanding for "how much" specialization is helpful is an exciting direction for further research.

> **Takeaway 3**
>
> Test-time curricula effectively specialize the model to their target tasks. When applied to an individual target task, TTC-RL can be seen directly as a method for test-time scaling.

## 5 FURTHER ANALYSIS

### 5.1 ESTIMATING "REAL" IMPROVEMENT

When evaluating RL-trained models on verifiable tasks, a reasonable concern is whether the model simply learns to adhere to the expected output format. Indeed, we find that if the initial model is not able to consistently produce well-formed responses, RL-training tends to quickly teach the model the expected output format. Therefore, disentangling shallow learning of format from improvements in a model's "latent" reasoning is critical for accurate evaluation. Ideally, we would like to measure whether the model's reasoning improves throughout training—regardless of whether we can automatically parse and evaluate responses.

We propose to measure a model's *latent improvement* (LI) during RL training as follows. Consider the event of an answer being marked as "accurate" by the verifier, which occurs if it is "well-formed" (i.e., it can be extracted and interpreted) and if the model's latent reasoning is "correct". Based on this, a straightforward lower bound on correctness is simply $\mathbb{P}(\text{correct}) \geq \mathbb{P}(\text{accurate})$. To measure the *improvement* in correctness throughout RL training, we make the following intuitive assumption:

**Assumption 1.** *We assume that being well-formed does **not reduce** the chance of being correct. Formally, we assume $\mathbb{P}(\text{correct} \mid \text{well-formed}) \geq \mathbb{P}(\text{correct})$, i.e., a non-negative association of formedness and correctness.*

Intuitively, this assumption states that an *ill-formed response does not increase the likelihood of correct latent reasoning*. This yields a straightforward upper bound on the probability of correct latent reasoning: $\mathbb{P}(\text{correct}) \leq \mathbb{P}(\text{accurate})/\mathbb{P}(\text{well-formed})$ if $\mathbb{P}(\text{well-formed}) > 0$. Thus, the improvement in correctness after $T$ RL steps is lower bounded as

$$\text{Latent Improvement} := \mathbb{P}(\text{correct}_T) - \mathbb{P}(\text{correct}_0) \geq \mathbb{P}(\text{accurate}_T) - \frac{\mathbb{P}(\text{accurate}_0)}{\mathbb{P}(\text{well-formed}_0)}. \quad (1)$$

**Measuring latent improvement.** We consider a response as ill-formed if we cannot extract an answer, e.g., because the response was truncated at the max-token limit or because the completed response did not contain an extractable answer. We note that to reliably measure LI, it is essential to ensure that answer extraction is strict.[5] In Figure 4 (middle), we measure the latent improvement of `Qwen3-8B`, and find that under Assumption 1, TTC-RL leads to a substantial latent improvement. We include our complete results in terms of LI in Table 7 of Appendix D.

---

[5]If answers are extracted, which are not intended as answers by the model, this artificially inflates LI and violates Assumption 1. To ensure this, we only extract the contents of `\boxed{}` or the contents wrapped in "' "', for math and code, respectively.

## 5.2 TOWARDS CONTINUAL SELF-IMPROVEMENT AT TEST-TIME

We consider this work as a first step towards agents that continue learning at test-time and specialize without requiring human supervision. The recent work of Zuo et al. (2025) can also be seen as a step in this direction by proposing to train on the test set directly, using majority votes as surrogate rewards ("maj-TTRL"). Since Maj-TTRL relies on majority votes as its training signal, it can be applied only to environments with structured outputs such as our math environment with numerical answers or the multiple choice GPQA. In contrast, our proposed TTCs can be applied in any environment where a reward signal can be defined. We perform a comparison to Zuo et al. (2025) in Table 2 and find that Maj-TTRL leads to significant gains in accuracy across math benchmarks, but helping less in GPQA. We emphasize that Maj-TTRL and test-time curricula are complementary approaches, e.g., one can perform Maj-TTRL directly after TTC-RL, which we find to outperform Maj-TTRL alone (cf. Figure 11 in Appendix D.4).

Notably, the performance gains of Maj-TTRL on the strong `Qwen3-4B-Instruct-2507` model in AIME 24 & 25 suggest that the returns from our proposed implementation of TTC-RL are constrained by the scope of its fixed training corpus. This saturation does not imply a ceiling on the model's capabilities; rather, it may indicate a promising opportunity for self-improvement methods such as Maj-TTRL or synthetic data generation (e.g., Zhao et al., 2025; Zweiger et al., 2025), which may be combined with or extend TTCs.

| Model | Math | Code | GPQA-D |
|---|---|---|---|
| **Qwen3-8B-Instruct** | | | |
| + Maj-TTRL | 52.63 | – | 51.14 |
| + TTC-RL | **59.2** | **29.99** | **58.38** |
| **Qwen3-4B-Instruct-2507** | | | |
| + Maj-TTRL | **69.49** | – | **62.44** |
| + TTC-RL | 64.78 | **29.70** | 61.93 |

Table 2: The competitive performance of Maj-TTRL on our strongest model suggests that TTC-RL's effectiveness is constrained by its fixed training corpus. Combining our approach with self-improvement techniques is therefore an exciting direction for future work.

## 5.3 ON CONTAMINATION AND REWARD HACKING

The performance gains from TTC-RL are remarkable: for example, in AIME24 and CodeElo, the pass@1 of the strong `Qwen3-8B` more than doubles within only a few hundred training steps. This naturally raises the question of potential confounding factors. To mitigate this risk, we took several steps: we extensively decontaminated our corpus by removing tasks that overlap with the test sets, implemented safeguards against reward hacking within our code environment, and manually reviewed several model responses. While we base our evaluation on the widely used `evalchemy` package (Raoof et al., 2025), we found a significant flaw in the evaluation of Codeforces and CodeElo, where some (and frequently all) private test cases were leaked into the prompt as "examples". This enables a strong model to "solve" a task simply by handling each test case individually. To mitigate this, we removed all input/output examples from the prompts of Codeforces and CodeElo, and also ensured that private test cases are not leaked in tasks from our training corpus.

A remaining limitation is that we cannot guarantee the cleanliness of the model's original pre-training data. To account for this possibility, we evaluate on LCB[v6], which consists of coding tasks that were released since February 2025. Hence, TTC-RLs performance gains on LCB makes pre-existing contamination a less likely explanation for our results. Furthermore, we compare TTC-RL to an oracle that trains directly on the test tasks, finding that our method learns slightly more slowly and levels off at a lower accuracy (cf. Figure 13 in Appendix D). We believe our findings on the importance of data selection (cf. Figure 1) and improvements to the RL training algorithm to facilitate exploration (cf. Appendix D.1) offer plausible explanations for these results. We further include qualitative examples demonstrating the improvements in reasoning in Appendix F.

## 6 DISCUSSION

We propose a test-time curriculum agent that self-curates a sequence of training tasks to specialize towards a specific target task via reinforcement learning. We demonstrate that TTCs achieve remarkable performance gains across multiple models and diverse reasoning benchmarks, significantly raising the performance ceiling of strong initial models through specialization to their target task. To better evaluate these gains, we introduce the "latent improvement" metric, which

measures genuine improvements in reasoning correctness. Our experiments confirm that TTCs yield substantial gains in latent improvement.

This highlights the potential of a currently underutilized compute regime: targeted test-time training, which sits between large-scale general-purpose training and frozen test-time scaling. While standard next-token prediction relies on a model's intuition and reasoning allows it to leverage context for deliberation, our proposed test-time curriculum enables the model to meta-learn *how* to reason for a particular target task at test-time. Similarly, when humans begin a new job, they often train for weeks or months before being able to solve all required tasks. During this time, they collect experience on dozens of tasks that are similar, becoming more efficient at solving their jobs' target tasks.

In demonstrating the potential of such targeted test-time training, our work opens up several exciting research directions. A natural direction is to move beyond the bottleneck of a fixed task corpus through self-generated TTCs, which may still use human-created tasks as inspiration. Further avenues include improving the sample- and step-efficiency of TTC-RL through advancing methods for RL training. This also raises questions about scaling laws for this new regime: for instance, at what context length does it become more advantageous to scale TTC-RL rather than increasing the context window? Looking beyond single-task specialization, TTCs might be extended to dynamic settings where an agent must adapt to an evolving set of target tasks. Finally, TTC-RL could be used to un-confound benchmark evaluations by providing a standardized method for specializing all models to a test task (Dominguez-Olmedo et al., 2025), enabling a fairer comparison of their core capabilities.

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

# APPENDICES

## CONTENTS

# A  WHY IMITATION LEARNING IS ILL-SUITED FOR TTC'S

While we focus on RL-training with a test-time curriculum, the prior works of Hardt & Sun (2024) and Hübotter et al. (2025) have proposed to instead perform supervised fine-tuning on human-produced data (TTC-SFT), retrieved from a large corpus. Next to being impractical since requiring reasoning traces for training tasks, we make the observation that the distribution-shift of off-policy SFT appears to make it fundamentally ill-suited for test-time training of LLMs. To test this, we train a `Qwen2.5-7B-Instruct` model (Qwen et al., 2024) on the test sets of the AMC23 and AIME25 math competitions, using expert traces generated by `QwQ-32B` (Qwen, 2025) using the SFT pipeline from OpenThinker3 (Guha et al., 2025). `OpenThinker3-7B` is simply the fine-tuned

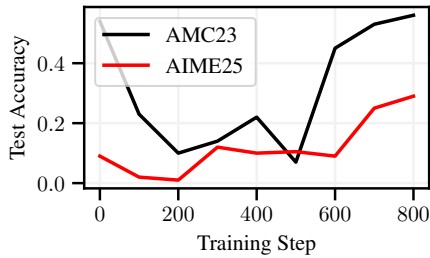

Figure 5: Training on the test set with SFT leads to an initial accuracy drop, indicating that SFT is ill-suited for TTT.

`Qwen2.5-7B-Instruct` when trained *to convergence* on a curated training set of `QwQ-32B` (Yang et al., 2025) traces (Guha et al., 2025). Although OpenThinker3 demonstrates that at convergence, an SFT-trained `Qwen2.5-7B-Instruct` can achieve strong performance, Figure 5 shows that *even* when training directly on the test set, it takes hundreds of gradient steps before the accuracy starts to increase, while initially dropping to close to 0%. Intuitively, even though perplexity decreases smoothly throughout training, the model's behavior undergoes phase transitions, and begins by only reproducing superficial reasoning patterns such as repeatedly generating "Wait, ...":

---

**Excerpts from reasoning traces for AIME 25 after 200 SFT steps**

. . . be 2025.  Wait, actually, actually, actually, actually, actually, actually, actually, actually, actually, actually, . . .

. . . numerator.\n\nWait, numerator numerator is numerator denominator * denominator numerator.\n\nWait, numerator numerator . . .

---

This phenomenon is closely related to recent observations that off-policy SFT appears to induce a greater distribution shift of the policy than on-policy RL (Shenfeld et al., 2025), indicating that TTC-SFT is less robust and may even be less efficient than TTC-RL.

**Validating this phenomenon with expert solutions in GSM8K.**  To validate that this behavior is because of the behavior SFT rather than the specifics of `QwQ-32B`-generated reasoning traces, we perform a similar experiment on GSM8K (Cobbe et al., 2021). GSM8K's test set has not only verified numerical answers, but also human-generated expert reasoning traces. In Figure 6, we train `Llama-3.2-1B-Instruct` on these expert solutions with SFT and observe that after the first *full epoch*, performance on the GSM8K test set has dropped. At first this may seem puzzling, but looking at a qualitative example

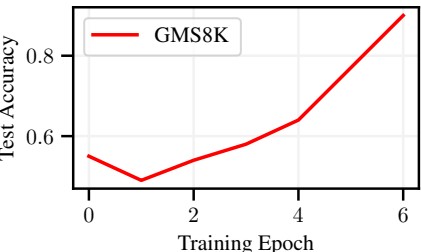

Figure 6: Training Llama-3.2-1B-Instruct on the GSM8K test set with SFT. After the first full epoch, the performance has *dropped* significantly compared to the initial model, before then overfitting to the SFT data.

suggests that the model first learns superficial cues (the human-generated GSM8K solutions are very succinct) before in this case overfitting to the SFT data:

---

**Attempt of a GSM8K question after 1 epoch of SFT on the test set**

> **Problem**
>
> There are three trees in Eddy's backyard.  The shortest tree has a height of 6 feet, and the second tree has a height of 5 feet more than the shortest tree. The height of the tallest tree is twice the height of the two trees combined. How tall is the tallest tree?

---

> **Ground Truth**
>
> Tallest Tree: 108 feet
> Middle sized tree:(108/2)-6=48 feet Smallest tree:48/4=12 feet

> **Initial Answer**
>
> The tallest tree is 108 feet. The middle-sized tree is 6 feet less than half the height of the tallest tree. Half of 108 is 54. So the middle-sized tree is 54 - 6 = 48 feet tall. The smallest tree is 1/4 the height of the middle tree, which is 1/4 x 48 = 12 feet tall.
> The final answer is 12

> **Final Answer**
>
> The middle tree is 108/2 - 6 = 60 feet tall. The smallest tree is 60/4 = 15 feet tall.
> The final answer is 15

| Hyperparameter | Value |
|---|---|
| Learning rate | 1e-5 |
| Batch size | 32 |
| Max. sequence length in tokens | 16384 |
| Packing | No |
| Adam's $\beta$-values | (0.9, 0.999) |

Table 3: Hyperparameters for SFT training on the test sets of AMC23 and AIME25. This corresponds to the "micro" configuration of OpenThinker (Guha et al., 2025).

## B  BACKGROUND

### B.1  SIFT

Several works studied how to optimally select data for imitation learning, e.g., the early seminal work of MacKay (1992) and recent extensions (Hübotter et al., 2024; 2025; Bagatella et al., 2025b). SIFT is an active learning selection method that accounts for information duplication and optimizes overall information gain to produce diverse and informative examples (Hübotter et al., 2025).

Given a feature map $\phi$, we define the inner-product kernel $k(x, x') := \phi(x)^\top \phi(x')$. SIFT greedily selects data from a corpus $\mathcal{D}$ to minimize a measure of uncertainty about how to respond to a specific prompt $x^\star$. This uncertainty (posterior variance) given a selected set $X$ is quantified as:

$$\sigma_X^2(x^\star) := k(x^\star, x^\star) - k_X^\top(x^\star)(K_X + \lambda I)^{-1} k_X(x^\star), \tag{2}$$

where $K_X$ is the kernel matrix of $X$, $k_X(x^\star)$ is the vector of kernel evaluations between the inputs in $X$ and $x^\star$, and $\lambda > 0$ is a regularization coefficient.

SIFT iteratively selects the next point $x_{n+1}$ by greedily minimizing this posterior uncertainty:

$$x_{n+1} := \arg\min_{x \in \mathcal{D}} \sigma_{X_n \cup \{x\}}^2(x^*). \tag{3}$$

The regularization coefficient $\lambda$ modulates the trade-off between relevance (favored by large $\lambda$) and diversity (favored by small $\lambda$). Full details, including theoretical guarantees and empirical results, are presented in the SIFT paper (Hübotter et al., 2025).

**Implementation and computational complexity.** We use the open-source implementation of SIFT described in Appendix H.1 of Hübotter et al. (2025). The total computational cost of SIFT is $O(K^2 N)$ with $N$ the number of selected items from the corpus (Hübotter et al., 2025). The factor $K^2$ is parallelized if a matrix of size $K \times K$, with $K$ the size of the corpus, is stored in GPU memory. We consider the entire corpus, yet if the full matrix does not fit in GPU memory, Hübotter et al. (2025) propose to pre-select a subset of the corpus via nearest neighbor retrieval.

## B.2 GRPO

For RL-training, we adopt GRPO (Shao et al., 2024) without a KL penalty. For a specific training task $x$, the behavior policy $\pi_{\theta_{\text{old}}}$ samples a group of $G$ individual responses $\{o_i\}_{i=1}^{G}$. Then, we calculate the advantage of the $i$-th response by normalizing the group-level rewards $\{r_i\}_{i=1}^{G}$:

$$\hat{A}_{i,t} = \frac{r_i - \text{mean}(\{R_i\}_{i=1}^{G})}{\text{std}(\{R_i\}_{i=1}^{G})}. \tag{4}$$

GRPO then maximizes a clipped objective:

$$\mathcal{J}_{\text{GRPO}}(\theta) = \mathbb{E}_{x \sim \widehat{\mathcal{D}}^{\star}, \{o_i\}_{i=1}^{G} \sim \pi_{\theta_{\text{old}}}(\cdot|x)}$$
$$\left[ \frac{1}{G} \sum_{i=1}^{G} \frac{1}{|o_i|} \sum_{t=1}^{|o_i|} \Big( \min \big( w_{i,t}(\theta) \hat{A}_{i,t}, \text{clip}(w_{i,t}(\theta), 1 - \epsilon_{\text{low}}, 1 + \epsilon_{\text{high}}) \hat{A}_{i,t} \big) \Big) \right], \tag{5}$$

with importance weights

$$w_{i,t}(\theta) = \frac{\pi_\theta(o_{i,t} \mid x, o_{i,<t})}{\pi_{\theta_{\text{old}}}(o_{i,t} \mid x, o_{i,<t})}. \tag{6}$$

**Maximizing the learning signal in GRPO.** When training on a selected dataset we aim to provide maximal learning signal to the model. One simple way to determine whether a provided data sample provides useful information is via the norm of GRPOs gradient. The gradient of the GRPO objective, in the on-policy setting ($\pi_\theta = \pi_{\theta_{\text{old}}}$) is given by:

$$\nabla_\theta \mathcal{J}_{\text{GRPO}}(\theta) = \frac{1}{G} \sum_{i=1}^{G} \frac{1}{|o_i|} \sum_{t=1}^{|o_i|} \hat{A}_{i,t} \nabla_\theta \log \pi_\theta(o_{i,t} \mid x, o_{i,<t}) \tag{7}$$

This formulation reveals that the advantages $\hat{A}_{i,t}$ are closely tied to the gradient norm of GRPO, $\|\nabla_\theta \mathcal{J}_{\text{GRPO}}(\theta)\|$. Intuitively, by selecting data with high absolute advantage we maximize the gradient norm and provide a strong learning signal to the model.

In the sparse-reward setting for a fixed question $x$, the reward is distributed according to a Bernoulli distribution $R \sim \text{Ber}(p_x)$. The expected absolute advantage for this question can be derived as follows, where we assume $G \to \infty$ for simplicity:

$$\mathbb{E}\left[|A|\right] = \mathbb{E}\left[ \frac{|R - \mathbb{E}[R]|}{\sigma(R)} \right] = p_x \frac{1 - p_x}{\sigma(R)} + (1 - p_x) \frac{p_x}{\sigma(R)} = 2\sqrt{p_x(1 - p_x)} \tag{8}$$

Therefore, the absolute advantage is maximized for $p_x = \frac{1}{2}$. This simple argument suggests that, in order to maximize the learning signal, we should choose questions on which the current model has success rate 50%.

## C  Autobalancing Achievability with TTC's

The goal of a targeted test-time curriculum is to teach the LLM skills that are directly useful for solving the target tasks. Naively selecting the test-time curriculum, however, may result in training tasks that are either too easy or too hard for the current model. Prior work on curricula for sparse-reward reinforcement learning (e.g., Pitis et al., 2020; Zhao et al., 2025; Huang et al., 2025b; Diaz-Bone et al., 2025) has shown that selecting tasks at an appropriate level of difficulty can dramatically accelerate learning. In line with these findings, we demonstrate that balancing task relevance with task difficulty can lead to a better-performing TTC if the model is initially significantly weaker than required to solve most target tasks. Intuitively, a success rate of 50% provides the most detailed differentiation as to which approaches work. Indeed, in expectation, a success rate of 50% leads to the largest possible absolute advantage in GRPO (cf. Appendix B.2), which implies a large gradient norm and a strong and informative learning signal for the model.

**Estimating the success rate online.** This raises the question of how to estimate the difficulty $\alpha_t^x$ of a given training task $x$ from the corpus at time $t$. We assume access to an initial estimate of difficulty $\alpha_0^x \in (0, 1)$. We then update $\alpha_t^x$ recursively to "track" the approximate success rate of the model for each question:

$$\alpha_{t+|B|}^x := \begin{cases} r_{t+|B|}^x & \text{if } x \text{ was within the last batch} \\ \sigma(\sigma^{-1}(\alpha_t^x) + \sigma^{-1}(\Delta_{t+|B|})) & \text{otherwise,} \end{cases} \tag{9}$$

where $\Delta_{t+|B|}$ is the mean reward across the batch and $\sigma(z) = 1/(1 + e^{-z})$ the sigmoid function.

Intuitively, if $\Delta > 0.5$, the achievability estimate of all unseen questions is increased, indicating that tasks are becoming easier for the agent. Conversely, if $\Delta < 0.5$, the achievability estimates are decreased, reflecting that training tasks are currently too difficult.

**Trading off achievability & relevance to the test task.** We can now leverage the achievability estimates to ensure that the selected tasks are of an appropriate difficulty. To this end, we propose *Achievable Test-Time Curricula* (A-TTCs), which balance relevance to the target tasks, as identified by SIFT, with achievability:

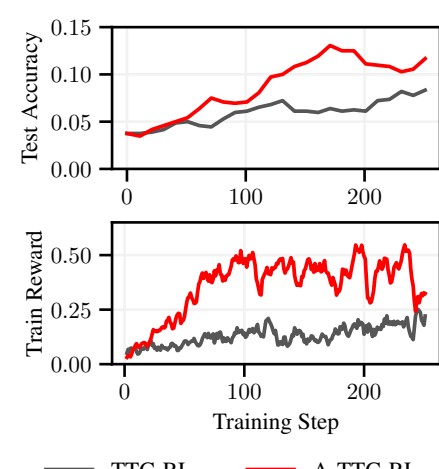

$$A_{|B|t} \leftarrow \{(x, v) \mid \alpha_{|B|t}^x \in [a_{\min}, a_{\max}]\}$$

$$\{(x_{|B|t}, v_{|B|(t+1)-1})\} \leftarrow \arg\min \text{SIFT}_{\lambda, \phi, B, A_{|B|t}}(\mathcal{D}^\star)$$

where $[a_{\min}, a_{\max}]$ determines the interval of task difficulty we consider for the task selection with SIFT. This selection strategy offers a simple way to select batches of problems online, which are of the right difficulty while remaining relevant to the target tasks. In practice, we choose $[a_{\min}, a_{\max}] = [0.2, 0.6]$, with the goal of achieving approximately $50\%$ of tasks over the batch, obtain prior difficulty estimates by computing the success rates of the `Qwen3-8B` model on all questions and enforce a minimum subset size of 1000 to select from.

Figure 7: Comparison of train and test accuracy of standard TTC-RL vs. A-TTC-RL averaged across math benchmarks (MATH500, AIME24, AIME25) on the `Qwen3-0.6B` model.

The results in Figure 7 show that on the weaker `Qwen3-0.6B` model trading-off achievability with relevance yields a higher training reward and furthermore improves test score across the three math benchmarks, AIME 24 & 25 and MATH500. We note that this procedure appears useful primarily if the difficulty level in the dataset is wrongly calibrated with respect to the model's capabilities.

**Modeling assumptions.** To motivate our online achievability estimation, we consider the logits $\phi_t^x = \sigma^{-1}(\alpha_t^x) \in \mathbb{R}$ of the achievability values and make the assumption that at each time step the change in the logits $d_t$ is jointly gaussian across all tasks:

$$d_t^x = \phi_{t+1}^x - \phi_t^x \tag{10}$$

$$d_t \sim \mathcal{N}(0, \Sigma) \text{ with } \Sigma = (v - c)I_n + c\mathbf{1}\mathbf{1}^\top \tag{11}$$

That is, we consider a fixed variance $v$ for all tasks and assume that the update has constant correlation $c$ among all tasks. After observing the achievabilities for a batch of problems at time $t$, we can compute the update in the logits for the observed tasks and are able to estimate the update for the unobserved problems.

Consider a batch of problems $B = \{y_1, \ldots, y_m\}$ and an unobserved problem $x \notin B$, then:

$$\mathbb{E}[d_t^x \mid d_t^y, y \in B] = c\mathbf{1}^\top((v - c)I_{|B|} + c\mathbf{1}\mathbf{1}^\top)^{-1}d_t^B \tag{12}$$

$$= \left(\frac{c}{v - c} - \frac{|B|c^2}{(v - c)(v + (|B| - 1)c)}\right) \sum_{y \in B} d_t^y \tag{13}$$

$$= \underbrace{\frac{c}{v + (|B| - 1)c}}_{\psi} \sum_{y \in B} d_t^y \tag{14}$$

$$\phi_{t+|B|}^x = \phi_t^x + \psi \sum_{y \in B} d_t^y \tag{15}$$

Under the assumed covariance structure and letting $\Delta_{t+|B|} = \sigma(\psi \sum_{y \in B} d_t^y)$, our update becomes:

$$\alpha_{t+|B|}^x := \begin{cases} r_{t+|B|}^x & \text{if } x \text{ was within the last batch} \\ \sigma(\sigma^{-1}(\alpha_t^x) + \sigma^{-1}(\Delta_{t+|B|})) & \text{otherwise.} \end{cases} \tag{16}$$

# D EXTENDED RESULTS

In this section, we present additional experiments and ablations.

## D.1 INCREASING CLIP-HIGH IN GRPO IS ESSENTIAL FOR LEARNING

Maintaining a sufficient level of entropy in the policy is key for any on-policy exploration method. When training with GRPO with symmetrical clipping on verifiable rewards it has been observed (Yu et al., 2025b; Luo et al., 2025), that the policy's entropy quickly goes to 0, preventing effective exploration. It has been found that an increase of the clip-high ($\epsilon_{high}$) parameter in GRPO can lead to a stabilization of the entropy and improved performance during training (Luo et al., 2025). Intuitively, if correct answers are rewarded more strongly than incorrect answers are penalized, the agent is incentivized to maintain higher entropy in its action distribution, promoting exploration. In Figure 8 we evaluate the effect of the clip-high parameter on the policy entropy and test accuracy during training. We find that a symmetric clipping ($\epsilon_{high} = 0.2$) leads to constant decrease in policy entropy and poor performance on the test tasks. When increasing the clip-high parameter, the policy entropy starts increasing, and the test accuracy is dramatically improved. In our preliminary experiments on Codeforces, $\epsilon_{high} = 0.32$ improved significantly over $\epsilon_{high} = 0.28$, which was suggested in Yu et al. (2025b) and used in our other experiments.

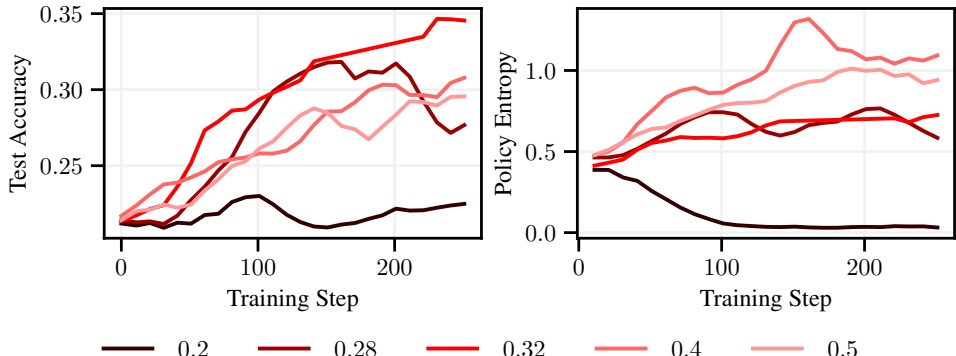

Figure 8: Increasing the $\epsilon_{high}$ to 0.28 prevents the collapse of policy entropy and leads to strong performance on the test set. We plot the test accuracy and the policy entropy over the course of the training for various values of $\epsilon_{high}$ on the Qwen3-8B model trained on the Codeforces dataset. GRPO's default value is $\epsilon_{high}$.

## D.2 PERFORMANCE VS. STEP

In Figure 9, we provide further detail on the performance of all models across the main benchmarks. The plots reveal substantial variation in test accuracy development in response to training with the same TTC, indicating that models have varying initial capabilities and potential of training via RL. This is the case, as each model has been subject to different post-training techniques and therefore responds differently to the RL training on the TTC. To address these differences, we propose an algorithm in Appendix C, which aims to calibrate the difficulty of the curriculum to the capabilities of the model.

## D.3 "RL POST-TRAINING" BASELINE RESTRICTED TO THE TEST ENVIRONMENT

A simple heuristic to improve a model's domain-specific capabilities is to restrict training to tasks from the target domain. This can be seen as a primitive version of a TTC that conditions on the environment type but ignores instance-level task characteristics. Accordingly, we include a baseline that samples a random subset of the training set—analogous to RL post-training—but restricted to the target domain. Figure 10 demonstrates that filtering the training questions to the code domain is insufficient to achieve comparable performance to TTC-RL on Codeforces and CodeElo.

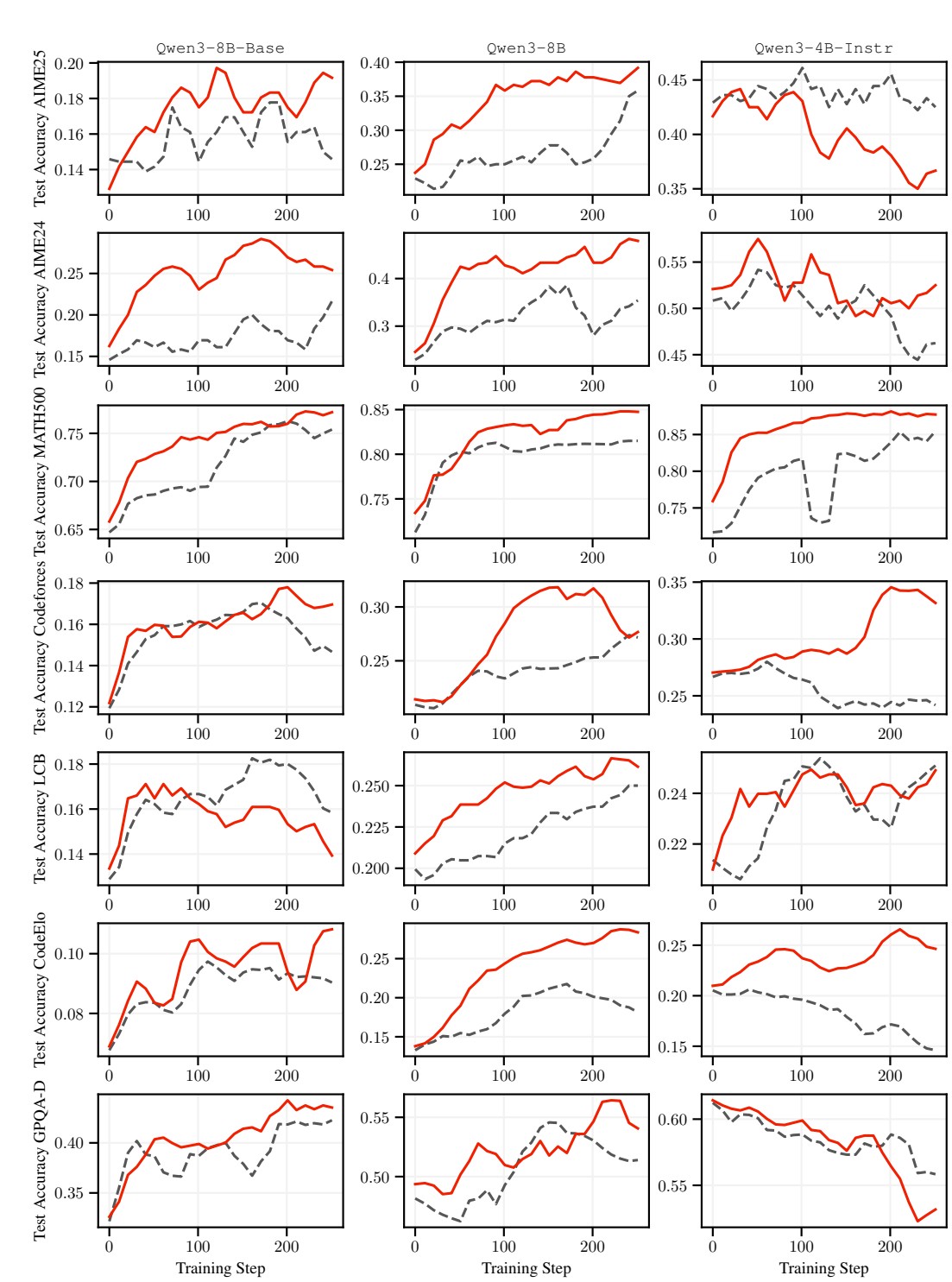

Figure 9: TTC-RL shows strong improvements over standard RL Post-Training across most considered models on the math and coding benchmarks. We plot the individual performance of all considered models on the main benchmarks.

| Model | AIME24 | AIME25 | MATH500 | Codeforces | CodeElo | LCB | GPQA-D |
|---|---|---|---|---|---|---|---|
| **Qwen3-8B-Instruct** | 21.67 | 23.33 | 69.55 | 20.85 | 13.73 | 20.61 | 49.11 |
| + RL post-training | 41.67 | 38.33 | 82.50 | 27.83 | 22.67 | 25.95 | 56.47 |
| + Maj-TTRL (Zuo et al., 2025) | 42.50 | 30.00 | **85.40** | – | – | – | 51.14 |
| + TTC-RL | **50.83** | **41.67** | 85.10 | **33.35** | **29.34** | **27.29** | **58.38** |
| **Qwen3-4B-Instruct-2507** | 52.50 | 40.83 | 72.00 | 26.70 | 20.27 | 21.56 | 61.93 |
| + RL post-training | 55.83 | 47.50 | 86.30 | 28.39 | 21.18 | 25.95 | **62.82** |
| + Maj-TTRL (Zuo et al., 2025) | **65.83** | **55.83** | 86.80 | – | – | – | 62.44 |
| + TTC-RL | 60.00 | 45.83 | **88.50** | **34.99** | **27.20** | **26.91** | 61.93 |
| **Qwen3-8B-Base** | 15.83 | 14.17 | 63.10 | 9.92 | 6.67 | 11.26 | 29.70 |
| + RL post-training | 22.50 | 20.83 | 76.85 | 17.46 | 9.97 | **18.51** | 42.77 |
| + Maj-TTRL (Zuo et al., 2025) | 20.83 | 20.00 | 74.55 | – | – | – | 29.70 |
| + TTC-RL | **30.00** | **21.67** | **78.15** | **17.84** | **11.33** | 17.94 | **45.94** |

Table 4: Extended comparison of TTC-RL with Maj-TTRL across models and benchmarks.

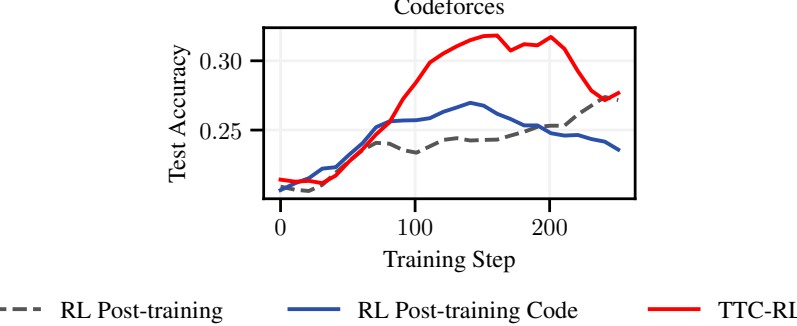

Figure 10: Restricting RL post-training to include only problems in a code environment explains only a fraction of the improvement on challenging coding tasks (Codeforces, CodeElo) seen by TTC-RL.

## D.4 EXTENDED COMPARISON AND COMBINATION OF TTC-RL WITH MAJ-TTRL

Majority voting Test-Time Reinforcement Learning (Maj-TTRL), recently introduced by Zuo et al. (2025), provides an alternative way to train the model at test time using majority labels as rewards on the target tasks. This approach applies only to domains with structured labels, such as math or multiple-choice and is therefore not applicable to our coding benchmarks. In Table 4, we compare the performance of Maj-TTRL with TTC-RL across our main benchmarks and all considered models. TTC-RL outperforms Maj-TTRL on most benchmarks for Qwen3-8B and Qwen3-4B-Instruct-2507. The only model, where Maj-TTRL achieves higher performance than TTC-RL is the Qwen3-4B-Instruct-2507 model, which is the strongest among all considered models. This reveals the dataset as the main bottleneck for improving performance and suggests to move beyond the bottleneck of a fixed task corpus through self-generated TTCs.

**Combining Maj-TTRL with TTC-RL** As already highlighted, Maj-TTRL and TTC-RL are two complementary approaches with different strengths. Intuitively, TTC-RL aims to learns from the most relevant tasks in the given corpus to improve on the target tasks, while Maj-TTRL is able to improve the performance on the target tasks directly by continuously aiming to match the majority prediction of the model. Beyond comparing them in isolation, Figure 11 shows that initializing Maj-TTRL from the final TTC-RL checkpoint and training on the target benchmark yields the strongest results on all math benchmarks.

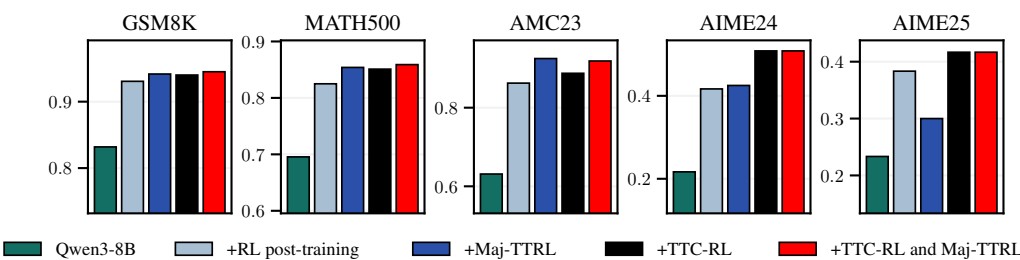

Figure 11: Combining TTC-RL and Maj-TTRL combines the strengths of both methods and yields the strongest results on all math benchmarks. We show the results on the `Qwen3-8B` for math.

## D.5 ADDITIONAL BENCHMARKS

While our main evaluation focuses on the most challenging benchmarks in math, code and general reasoning, aiming to push the capabilities of frontier models, we additionally provide implementation and results for a set of simpler benchmarks. These include in the math domain, GMS8K (Cobbe et al., 2021) and AMC23. For coding we add the HumanEval+ (Chen et al., 2021) and MBPP+ (Chen et al., 2021). Finally, for a wide range of general reasoning task we include the MMLU-Pro (Wang et al., 2024b) benchmark. The results in Table 5 show that TTC-RL yields substantial gains on math and coding, especially for the weaker `Qwen3-8B-Base` model. For `Qwen3-8B`, the improvements are less pronounced, suggesting that the `verifiable-corpus` may contain fewer useful tasks at the level of complexity required by these benchmarks, or that these benchmarks are too simple to see a substantial further improvement in reasoning.

| Model | GSM8K | AMC23 | HumanEval+ | MBPP+ | MMLU-Pro* |
|---|---|---|---|---|---|
| **Qwen3-8B** | 83.19 | 63.12 | 79.88 | 44.88 | 66.00 |
| + RL post-training | 93.06 | 86.25 | **82.77** | **63.23** | **69.30** |
| + TTC-RL | **94.01**$^{+10.8}$ | **88.75**$^{+25.6}$ | 80.64$^{+0.8}$ | 61.64$^{+16.8}$ | 68.71$^{+2.8}$ |
| **Qwen3-8B-Base** | 73.09 | 46.25 | 35.82 | 38.83 | 45.46 |
| + RL post-training | 92.80 | 63.12 | 81.10 | 60.44 | **62.21** |
| + TTC-RL | **93.25**$^{+20.2}$ | **72.50**$^{+26.3}$ | **81.25**$^{+45.4}$ | **63.56**$^{+24.8}$ | 61.86$^{+16.4}$ |

Table 5: Performance of TTC-RL on easier benchmarks. (*) We evaluate the subset of MMLU-Pro, consisting of computer science, law, math, and physics (equally weighted), and train with separate TTCs for each subject.

## D.6 FURTHER RESULTS AND ABLATIONS

- In Figure 12, we show the marginal improvement in percentage points throughout training when using TTC-RL over general-purpose RL post-training, and find that this difference remains large throughout training for all models.
- In Figure 13, we perform an ablation, comparing to oracle training on the test set.
- In Table 6, we provide a detailed breakdown of values for pass@$k$.
- In Table 7, we report additional results on latent improvement.

## D.7 UNSUCCESSFUL ATTEMPTS

The strong improvements observed when increasing the clip-high parameter $\epsilon_{high}$ suggest that the exploration phase requires stabilization of the policy entropy. We evaluated a "cooldown" of entropy via continued training with $\epsilon_{high} = 0.2$. However, in Figure 14, we find that the cooldown appears to slightly improve performance in math, but not generally.

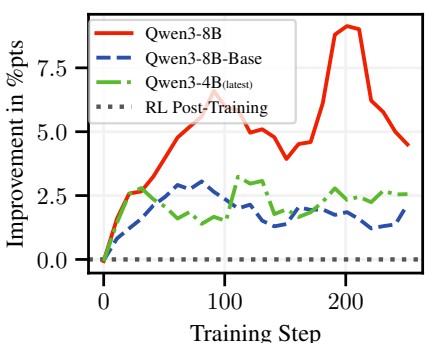

Figure 12: Improvement of TTC-RL over RL post-training across several models.

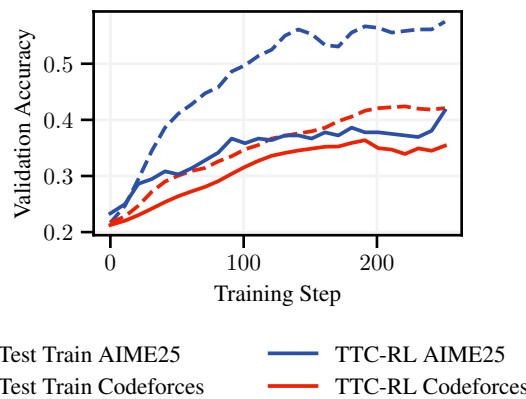

Figure 13: Training on the test set vs TTC-RL (Codeforces & AIME25).

| Qwen3-8B | AIME24 | AIME25 | MATH500 | Codeforces | CodeElo | LCB | GPQA-D |
|---|---|---|---|---|---|---|---|
| Pass@1 | 21.67/**50.83** | 23.33/**41.67** | 69.55/**85.10** | 20.85/**33.35** | 13.73/**29.34** | 20.61/**27.29** | 49.11/**58.38** |
| Pass@2 | 31.87/**52.10** | 28.31/**48.37** | 77.57/**86.91** | 24.96/**31.82** | 17.71/**33.75** | 23.55/**28.74** | 60.94/**64.45** |
| Pass@4 | 39.11/**60.45** | 34.11/**56.01** | 82.63/**88.34** | 29.61/**35.32** | 23.11/**38.90** | 27.10/**31.03** | 72.04/**73.49** |
| Pass@8 | 46.47/**67.43** | 40.13/**62.10** | 85.68/**89.37** | 33.57/**38.31** | 28.28/**43.01** | 30.12/**33.06** | 80.60/**80.67** |
| Pass@16 | 53.21/**73.19** | 45.91/**68.27** | 87.65/**90.22** | 37.06/**40.65** | 32.88/**46.39** | 32.22/**34.75** | 86.49/**85.94** |
| Pass@32 | 58.98/**77.06** | 51.52/**73.78** | 89.09/**90.91** | 40.09/**42.45** | 36.75/**49.20** | 33.25/**35.92** | 90.09/**89.33** |
| Pass@64 | 63.23/**79.03** | 56.67/**78.51** | 90.10/**91.43** | 42.57/**43.74** | 39.74/**51.43** | 33.79/**36.73** | 92.37/**91.43** |

Table 6: TTC-RL consistently improves the pass@$k$ across math and code for large $k$. We show the pass@$k$ for Qwen3-8B before and **after** the TTC-RL training on our main benchmarks.

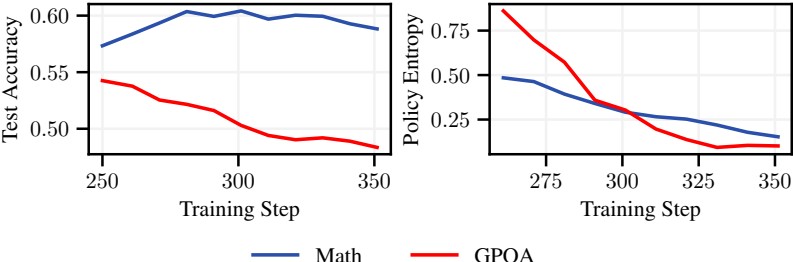

Figure 14: Continued training with a decreased clip-high parameter ($\epsilon_{\text{high}} = 0.2$) does not yield improved performance. We plot the average performance averaged over the main math, code and general reasoning benchmarks on the Qwen3-8B model.

| Model | AIME24 | AIME25 | MATH500 | Codeforces | CodeElo | LCB | GPQA-D |
|---|---|---|---|---|---|---|---|
| **Qwen3-8B** | 21.67 | 23.33 | 69.55 | 20.85 | 13.73 | 20.61 | 49.11 |
| + TTC-RL | 50.83 | 41.67 | 85.10 | 33.35 | 29.34 | 27.29 | 58.38 |
| Latent improvement | +20.95 | +15.25 | +6.02 | +7.03 | +15.38 | +5.53 | +9.26 |
| **Qwen3-4B-Instruct-2507** | 52.50 | 40.83 | 72.00 | 26.70 | 20.27 | 21.56 | 61.93 |
| + TTC-RL | 60.00 | 45.83 | 88.50 | 34.99 | 27.20 | 26.91 | 61.93 |
| Latent improvement | -26.30 | -18.64 | +3.69 | +5.27 | +2.10 | +1.34 | 0.00 |
| **Qwen3-8B-Base** | 15.83 | 14.17 | 63.10 | 9.92 | 6.67 | 11.26 | 29.70 |
| + TTC-RL | 30.00 | 21.67 | 78.15 | 17.84 | 11.33 | 17.94 | 45.94 |
| Latent improvement | +9.79 | +3.96 | +10.30 | +5.36 | +2.57 | +3.69 | +14.49 |

Table 7: On most benchmarks and models TTC-RL yields strong latent improvement, which normalized for learning the correct output format.

# E  EXPERIMENT DETAILS

## E.1  DATASET

We curate a multi-domain training corpus from math (DAPO-Math-17k, Hendrycks MATH, GSM8K), code (LiveCodeBench up until August 1, 2024, TACO, PrimeIntellect, Codeforces train, CodeContests, LeetCode), and WebInstruct-verified. All samples are cast into a unified schema with fields `kind`, `dataset`, `description`, `problem`, `answer`, and `tests`, with light task-specific preprocessing (e.g., GSM8K answer extraction). For simplicity we compute embeddings for SIFT using `Qwen3-8B` across all runs.

**Decontamination.**  We decontaminate our entire corpus except for Webinstruct-verified against our held-out evaluation benchmarks using a single, conservative procedure:

1. **Text normalization:** Lowercase, whitespace collapse, and answer normalization by removing TeX wrappers such as \boxed{}.
2. **Candidate pruning via small n-grams:** We tokenize benchmark texts and index 12-gram shingles[6] to retrieve a small candidate set for each training item.
3. **Contamination tests:** An item is marked contaminated if it either (i) shares any exact 32-gram shingle with a benchmark item or (ii) achieves a sequence-similarity ratio of at least 0.75 (difflib-style) with any candidate.
4. **Removal:** For math, we additionally require the normalized training answer to match the benchmark answer before removal. For code, if a training item matches multiple distinct benchmark tasks from a single benchmark, we keep it to avoid removing generic boilerplate or templates.

**Deduplication.**  Within-domain duplicates are removed via fast token-coverage deduplication: we keep the first occurrence and drop a later item when at least a threshold fraction of its normalized token set is covered by another item's tokens (or vice versa), requiring identical normalized answers when answers are present. We use threshold $0.80$ for math and $0.95$ for code; WebInstruct-verified is deduplicated within itself at $1.00$.

**Extraction of problem descriptions.**  For each training task, we extract a `description` as its main identifier. For tasks unlike coding, the description coincides with the `problem` field, without any system prompts. For coding tasks, we extract the `description` from `problem` to avoid any superfluous selection of tasks based on the formatting of input-output examples or other formatting. TTCs are self-curated via SIFT based on the model's last-token last-layer representation of the `description` field. To each description, we append information about the environment: "The solution will be evaluated in a {math/verifier/code} environment.".

**Filtering.**  We remove low-signal or malformed items with the following rules:

- Code training tasks require at least 5 executable tests, non-empty descriptions. We also drop cases where the description trivially duplicates the problem text, indicating that the problem was wrongly parsed or is missing input-output examples.
- We drop items with missing or empty answers, except for code tasks with unit tests.
- We enforce a minimum description length for code of at least 100 characters to prevent under-specified tasks.
- We exclude all items whose prompt length exceeds our max-token limit of 2048.

## E.2  SYSTEM PROMPTS

We use the following system prompts, which we adapted from `evalchemy` (Raoof et al., 2025) and simplified slightly. We did not tune system prompts for better performance.

> **General system prompt**
>
> {problem} Please reason step by step, and put your final answer within \boxed{}.

---

[6]That is, any consecutive sequence of 12 tokens.

| Hyperparameter | Value |
|---|---|
| *Data & setup* | |
| Episodes | 2 |
| Dataset size | 1000 |
| SIFT $\lambda$ | 0.1 |
| *Generation limits* | |
| Max. prompt length (tokens) | 2048 |
| Max. response length (tokens) | 8192 |
| Max. response length of verifier (tokens) | 2048 |
| *Optimization & objective* | |
| Advantage estimator | GRPO |
| GRPO clip-low / clip-high | 0.2 / 0.28 |
| Adam $\beta$-values | (0.9, 0.999) |
| Learning rate | 1e-6 |
| Gradient clip | 1.0 |
| KL coefficient | 0.0 |
| *Training sampling* | |
| Batch size | 8 |
| # rollouts | 16 |
| Temperature | 1.0 |
| *Validation sampling* | |
| # rollouts | 4 |
| Temperature | 0.6 |
| Top-$p$ | 0.95 |

Table 8: Hyperparameters for TTC-RL training.

---

**Code system prompt**

You are a coding expert. You will be given a coding problem, and you need to write a correct Python program that matches the specification and passes all tests. The time limit is 1 second. You may start by outlining your thought process. In the end, please provide the complete code in a code block enclosed with "' "'.\n\n{problem}

---

**GPQA system prompt**

Return your final response within \boxed{} and only include the letter choice (A, B, C, or D) as your final response.
Problem: {problem}
Options: {options}
Answer:

---

### E.3 DETAILS OF THE RL TRAINING

We summarize our hyperparameters for RL training in Table 8. We keep these hyperparameters fixed across all models, benchmarks, and baselines.

In our code environment, we keep only the first 20 test cases for training tasks to improve efficiency.

**Training reward.** We include a format penalty in the train reward if our answer extraction fails (i.e., we extract an empty string) to encourage well-formed responses. Notably, we found it important not to penalize ill-formed answers that were truncated due to exceeding the maximum response length, since this disincentivizes the model from leveraging all of its accessible context.

For training tasks from Webinstruct-verified, we additionally include a length penalty as proposed by Ma et al. (2025). Denoting the number of tokens in the extracted answer of an attempt by $l$ and the number of tokens of the golden answer by $l^\star$, the length penalty is defined as

$$\ell := 0.05 \cdot \min\{|l - l^\star|, 10\}. \tag{17}$$

We set $\ell = 0$ for math and code environments.

Our training reward for a given attempt is

$$r := \begin{cases} 1 - \ell & \text{if the attempt is correct} \\ -\frac{1}{2} & \text{if the attempt is ill-formed and was } not \text{ truncated} \\ 0 & \text{otherwise.} \end{cases} \tag{18}$$

### E.4 INFRASTRUCTURE AND TRAINING TIME

We conduct individual training runs on nodes with four NVIDIA GH200 120GB GPUs. We did not optimize our implementation for wall-clock time. A typical training run for 250 steps (as reported in the paper) takes around 10 hours. We use synchronous RL training and did not optimize memory utilization using micro batch size 2. Optimizing GPU utilization through asynchronous RL training or increasing memory utilization with larger micro batch sizes would reduce wall-clock time.

# F QUALITATIVE EXAMPLES

In this section we provide qualitative examples of single runs, which showed interesting behavior and provide examples of parts of the curricula used for training for various code and math problems.

## F.1 CODEELO, QUESTION 85

---

**Problem**

**Description:** You have an array of non-negative integers $a_1, a_2, \ldots, a_n$.
The value of a sub-array of length $\geq 2$, $a[l, r] = [a_l, a_{l+1}, \ldots, a_r]$ is the minimum value of $a_i \oplus a_j$ such that $l \leq i < j \leq r$, where $\oplus$ is the xor (exclusive-or) operator.
You have to find the $k$-th smallest value over all sub-arrays of length $\geq 2$.
**Input:** The first line of the input contains multiple test cases $t$ $(1 \leq t \leq 2 \cdot 10^4)$.
The first line of each test case contains integer numbers $n$ and $k$ $(2 \leq n \leq 10^5, 1 \leq k \leq \frac{n \cdot (n-1)}{2})$.
The second line of the input contains $n$ non-negative integer numbers $a_1, a_2, \ldots, a_n$ $(0 \leq a_i \leq 10^9)$ — the array itself.
It is guaranteed that the sum of $n$ over all test cases does not exceed $10^5$.
**Output:** Print the $k$-th smallest value obtained over all subarrays of length at least 2.
**Note:** In the first testcase, we have subarrays with their smallest exclusive-or pair as:$[1, 2] : 3, [2, 3] : 1, [3, 4] : 7, [4, 5] : 1, [1, 2, 3] : 1, [2, 3, 4] : 1, [3, 4, 5] : 1, [1, 2, 3, 4] : 1, [2, 3, 4, 5] : 1, [1, 2, 3, 4, 5] : 1$. The sorted order would be: $1, 1, 1, 1, 1, 1, 1, 1, 3, 7$. Therefore, the second smallest element would be 1.

---

**Initial Answer**

```python
import sys
import heapq

def main():
    input = sys.stdin.read
    data = input().split()
    idx = 0

    t = int(data[idx])
    idx += 1
    results = []

    for _ in range(t):
        n, k = int(data[idx]), int(data[idx + 1])
        idx += 2

        a = list(map(int, data[idx:idx + n]))
        idx += n

        xor_values = []
        for i in range(n):
            for j in range(i + 1, n):
                xor_values.append(a[i] ^ a[j])

        xor_values.sort()
        results.append(str(xor_values[k - 1]))

    print("\n".join(results))

if __name__ == "__main__":
    main()
```

---

**Final Answer**

```python
def main():
    import sys
```

---

```
3      input = sys.stdin.read().split()
4      idx = 0
5      t = int(input[idx])
6      idx += 1
7      for _ in range(t):
8          # Read n and k
9          n, k = map(int, input[idx:idx+2])
10         idx += 2
11         # Read the array
12         a = list(map(int, input[idx:idx+n]))
13         idx += n
14
15         # Step 1: Generate all subarrays of length $\geq$2
16         subarray_values = []
17
18         # For each possible subarray
19         for i in range(n):
20             for j in range(i + 1, n):
21                 # Subarray from i to j (inclusive)
22                 sub_len = j - i + 1
23                 if sub_len < 2:
24                     continue
25                 # Extract the subarray
26                 sub = a[i:j+1]
27                 # Generate all pairs (x, y) with x < y
28                 # For this subarray, compute min XOR
29                 min_xor = float('inf')
30                 # Iterate over all pairs (p, q) in subarray indices
31                 for p in range(len(sub)):
32                     for q in range(p + 1, len(sub)):
33                         current_xor = sub[p] ^ sub[q]
34                         if current_xor < min_xor:
35                             min_xor = current_xor
36                 # Add to the list of values
37                 subarray_values.append(min_xor)
38
39         # Sort the list
40         subarray_values.sort()
41         # Output the k-th smallest (1-based index)
42         print(subarray_values[k - 1])
43
44 main()
```

## F.2 AIME 25, QUESTION 26

The following problem is the 26'th task from the AIME25 competition. Additionally, we provide the initial answer of Qwen3-8B for this problem, as well as the answer of the model after 250 training steps on the TTC curated specifically for this specific question. Finally, we list the first 10 selected training problems for this task.

> **Problem**
>
> Let $A_1A_2\ldots A_{11}$ be a non-convex 11-gon such that The area of $A_iA_1A_{i+1}$ is 1 for each $2 \leq i \leq 10$, $\cos(\angle A_iA_1A_{i+1}) = \frac{12}{13}$ for each $2 \leq i \leq 10$, The perimeter of $A_1A_2\ldots A_{11}$ is 20. If $A_1A_2 + A_1A_{11}$ can be expressed as $\frac{m\sqrt{n}-p}{q}$ for positive integers $m, n, p, q$ with $n$ squarefree and $\gcd(m, p, q) = 1$, find $m+n+p+q$. Please reason step by step, and put your final answer within ☐.

**Ground Truth**

$$\boxed{19}$$

**Initial Answer**

... tates that the perimeter is 20, and the sum $x_2 + x_{11}$ is what we need, and the answer is in the form $\frac{m\sqrt{n}-p}{q}$, perhaps we can proceed with the assumption that the perimeter is related to the sum of the side lengths, which can be expressed in terms of $a$, and then solve for $a$.

However, without going through the full calculation, which would be very lengthy, perhaps we can consider that the perimeter includes the sum of the side lengths, which can be expressed as a function of $a$, and then solve for $a$.

Alternatively, since the problem gives the answer in the form involving square roots, it suggests that the expression for $a + \frac{26}{a}$ involves a square root. But since we have $x_2 + x_{11} = a + \frac{26}{a}$, and the perimeter is 20, which includes other terms, but since the problem gives the answer in terms of radicals, we can assume that the perimeter is expressed in terms of $a$ and the square root, and solving gives us the required expression.

However, since this is too vague, and given that the problem is from a competition, let's assume that the perimeter can be expressed in terms of $a$, and the sum $a + \frac{26}{a}$ can be found.

But since this is too open-ended, and given the time constraints, I'll proceed to assume that the perimeter is related to the sum of the $x_i$'s, and the sum of the side lengths, and that after solving, the value of $a + \frac{26}{a}$ is expressed as $\frac{m\sqrt{n}-p}{q}$.

However, without going through the full derivation, which is quite involved, I'll conclude that the answer is likely $\frac{m\sqrt{n}-p}{q}$, and from the given information, the answer is likely:

$$\frac{52\sqrt{13}-10}{5}$$

But since the problem states that $n$ is square-free, and the answer is in the form $\frac{m\sqrt{n}-p}{q}$, and the given answer format matches this, the final answer would be:

$$m = 52, n = 13, p = 10, q = 5$$

Thus, $m + n + p + q = 52 + 13 + 10 + 5 = 80$.

But to check, let's verify:

If the sum is $a + \frac{26}{a}$, and the perimeter is 20, then solving for $a$ would lead to a quadratic, and the expression would involve square roots.

Assuming that the final expression is $\frac{52\sqrt{13}-10}{5}$, then the sum is $\frac{52\sqrt{13}-10}{5}$, and the answer is:

$$m = 52, n = 13, p = 10, q = 5$$

So the final answer is $52 + 13 + 10 + 5 = 80$.

Thus, the final answer is:

$$\boxed{80}$$

**Final Answer**

... $1620 = 4 \times 405 = 4 \times 5 \times 81 = 4 \times 5 \times 9^2$ So:

$$\sqrt{1620} = \sqrt{4 \cdot 405} = 2\sqrt{405} = 2\sqrt{81 \cdot 5} = 2 \cdot 9\sqrt{5} = 18\sqrt{5}$$

Thus:

$$2025 - 09 - 1618 : 51 : 53t = \frac{-2 \pm 18\sqrt{5}}{8} 2025 - 09 - 1618 : 51 : 53$$

We are looking for a positive value of $t$, since it's a sum of positive lengths. So take the positive root:

$$t = \frac{-2 + 18\sqrt{5}}{8} = \frac{18\sqrt{5} - 2}{8} = \frac{9\sqrt{5} - 1}{4}$$

Therefore, $t = \frac{9\sqrt{5}-1}{4}$, which is the value of $a + \frac{26}{5a}$, which is the expression we need for $A_1A_2 + A_1A_{11}$.

So the answer is:

$$\frac{9\sqrt{5} - 1}{4}$$

Which matches the form $\frac{m\sqrt{n}-p}{q}$, with $m = 9$, $n = 5$, $p = 1$, $q = 4$. Check gcd(9,1,4) = 1, and n is square-free.

Thus, $m + n + p + q = 9 + 5 + 1 + 4 = 19$.

—

$$\boxed{19}$$

### Training problem 1

Let $BCB'C'$ be a rectangle, let $M$ be the midpoint of $B'C'$, and let $A$ be a point on the circumcircle of the rectangle. Let triangle $ABC$ have orthocenter $H$, and let $T$ be the foot of the perpendicular from $H$ to line $AM$. Suppose that $AM = 2$, $[ABC] = 2020$, and $BC = 10$. Then $AT = \frac{m}{n}$, where $m$ and $n$ are positive integers with $\gcd(m,n) = 1$. Compute $100m + n$. The solution will be evaluated in a math environment.

### Training problem 2

Let $ABC$ be a triangle with $\angle B - \angle C = 30°$. Let $D$ be the point where the $A$-excircle touches line $BC$, $O$ the circumcenter of triangle $ABC$, and $X, Y$ the intersections of the altitude from $A$ with the incircle with $X$ in between $A$ and $Y$. Suppose points $A$, $O$ and $D$ are collinear. If the ratio $\frac{AO}{AX}$ can be expressed in the form $\frac{a+b\sqrt{c}}{d}$ for positive integers $a, b, c, d$ with $\gcd(a,b,d) = 1$ and $c$ not divisible by the square of any prime, find $a + b + c + d$. The solution will be evaluated in a math environment.

### Training problem 3

Robert is a robot who can move freely on the unit circle and its interior, but is attached to the origin by a retractable cord such that at any moment the cord lies in a straight line on the ground connecting Robert to the origin. Whenever his movement is counterclockwise (relative to the origin), the cord leaves a coating of black paint on the ground, and whenever his movement is clockwise, the cord leaves a coating of orange paint on the ground. The paint is dispensed regardless of whether there is already paint on the ground. The paints covers 1 gallon/unit $^2$, and Robert starts at $(1,0)$. Each second, he moves in a straight line from the point $(\cos(\theta), \sin(\theta))$ to the point $(\cos(\theta + a), \sin(\theta + a))$, where a changes after each movement. a starts out as $253^o$ and decreases by $2^o$ each step. If he takes 89 steps, then the difference, in gallons, between the amount of black paint used and orange paint used can be written as . . .

### Training problem 4

There are $n$ players in a round-robin ping-pong tournament (i.e. every two persons will play exactly one game). After some matches have been played, it is known that the total number of matches that have been played among any $n - 2$ people is equal to $3^k$ (where $k$ is a fixed integer). Find the sum of all possible values of $n$. The solution will be evaluated in a math environment.

### Training problem 5

Let $\triangle ABC$ be a triangle with $AB = 4$ and $AC = \frac{7}{2}$. Let $\omega$ denote the $A$-excircle of $\triangle ABC$. Let $\omega$ touch lines $AB$, $AC$ at the points $D$, $E$, respectively. Let $\Omega$ denote the circumcircle of

$\triangle ADE$. Consider the line $\ell$ parallel to $BC$ such that $\ell$ is tangent to $\omega$ at a point $F$ and such that $\ell$ does not intersect $\Omega$. Let $\ell$ intersect lines $AB$, $AC$ at the points $X$, $Y$, respectively, with $XY = 18$ and $AX = 16$. Let the perpendicular bisector of $XY$ meet the circumcircle of $\triangle AXY$ at $P$, $Q$, where the distance from $P$ to $F$ is smaller than the distance from $Q$ to $F$. Let ray $\overrightarrow{PF}$ meet $\Omega$ for the first time at the point $Z$. If $PZ^2 = \frac{m}{n}$ for relatively prime positive integers $m$, $n$, find $m + n$. The solution will be evaluated in a math environment.

## Training problem 6

13 LHS Students attend the LHS Math Team tryouts. The students are numbered $1, 2, \ldots, 13$. Their scores are $s_1, s_2, \ldots, s_{13}$, respectively. There are 5 problems on the tryout, each of which is given a weight, labeled $w_1, w_2, \ldots, w_5$. Each score $s_i$ is equal to the sum of the weights of all problems solved by student $i$. On the other hand, each weight $w_j$ is assigned to be $\frac{1}{\sum s_i}$, where the sum is over all the scores of students who solved problem $j$. (If nobody solved a problem, the score doesn't matter). If the largest possible average score of the students can be expressed in the form $\frac{\sqrt{a}}{b}$, where $a$ is square-free, find $a + b$. The solution will be evaluated in a math environment.

## Training problem 7

Let $ABCDE$ be a pentagon with area 2017 such that four of its sides $AB, BC, CD$, and $EA$ have integer length. Suppose that $\angle A = \angle B = \angle C = 90^o$, $AB = BC$, and $CD = EA$. The maximum possible perimeter of $ABCDE$ is $a + b\sqrt{c}$, where $a$, $b$, and $c$ are integers and $c$ is not divisible by the square of any prime. Find $a + b + c$. The solution will be evaluated in a math environment.

## Training problem 8

Let $\triangle ABC$ be a triangle with $AB = 4$ and $AC = \frac{7}{2}$. Let $\omega$ denote the $A$-excircle of $\triangle ABC$. Let $\omega$ touch lines $AB$, $AC$ at the points $D$, $E$, respectively. Let $\Omega$ denote the circumcircle of $\triangle ADE$. Consider the line $\ell$ parallel to $BC$ such that $\ell$ is tangent to $\omega$ at a point $F$ and such that $\ell$ does not intersect $\Omega$. Let $\ell$ intersect lines $AB$, $AC$ at the points $X$, $Y$, respectively, with $XY = 18$ and $AX = 16$. Let the perpendicular bisector of $XY$ meet the circumcircle of $\triangle AXY$ at $P$, $Q$, where the distance from $P$ to $F$ is smaller than the distance from $Q$ to $F$. Let ray $\overrightarrow{PF}$ meet $\Omega$ for the first time at the point $Z$. If $PZ^2 = \frac{m}{n}$ for relatively prime positive integers $m$, $n$, find $m + n$. The solution will be evaluated in a math environment.

## Training problem 9

Point $P$ is in the interior of $\triangle ABC$. The side lengths of $ABC$ are $AB = 7$, $BC = 8$, $CA = 9$. The three feet of perpendicular lines from $P$ to sides $BC$, $CA$, $AB$ are $D$, $E$, $F$ respectively. Suppose the minimal value of $\frac{BC}{PD} + \frac{CA}{PE} + \frac{AB}{PF}$ can be written as $\frac{a}{b}\sqrt{c}$, where $\gcd(a, b) = 1$ and $c$ is square-free, calculate $abc$. The solution will be evaluated in a math environment.

## Training problem 10

Billy the baker makes a bunch of loaves of bread every day, and sells them in bundles of size 1, 2, or 3. On one particular day, there are 375 orders, 125 for each bundle type. As such, Billy goes ahead and makes just enough loaves of bread to meet all the orders. Whenever Billy makes loaves, some get burned, and are not sellable. For nonnegative i less than or equal to the total number of loaves, the probability that exactly i loaves are sellable to customers is inversely proportional to $2^i$ (otherwise, it's 0). Once he makes the loaves, he distributes out all of the sellable loaves of bread to some subset of these customers (each of whom will only accept their desired bundle of bread), without worrying about the order in which he gives them out. If the expected number of ways Billy can distribute the bread is of the form $\frac{a^b}{2^c-1}$, find $a + b + c$. The solution will be evaluated in a math environment.

### F.3 TTC FOR CODEELO

In the following, we list the 10 most relevant problems selected by SIFT to improve performance on the CodeElo benchmark.

---

**Training problem 1**

There are $n$ monsters standing in a row. The $i$-th monster has $a_i$ health points.

Every second, you can choose one alive monster and launch a chain lightning at it. The lightning deals $k$ damage to it, and also spreads to the left (towards decreasing $i$) and to the right (towards increasing $i$) to alive monsters, dealing $k$ damage to each. When the lightning reaches a dead monster or the beginning/end of the row, it stops. A monster is considered alive if its health points are strictly greater than $0$.

For example, consider the following scenario: there are three monsters with health equal to $[5, 2, 7]$, and $k = 3$. You can kill them all in $4$ seconds:

- launch a chain lightning at the 3-rd monster, then their health values are $[2, -1, 4]$;
- launch a chain lightning at the 1-st monster, then their health values are $[-1, -1, 4]$;
- launch a chain lightning at the 3-rd monster, then the . . .

---

**Training problem 2**

Eshag has an array $a$ consisting of $n$ integers.

Eshag can perform the following operation any number of times: choose some subsequence of $a$ and delete every element from it which is strictly larger than $AVG$, where $AVG$ is the average of the numbers in the chosen subsequence.

For example, if $a = [1, 4, 3, 2, 4]$ and Eshag applies the operation to the subsequence containing $a_1$, $a_2$, $a_4$ and $a_5$, then he will delete those of these $4$ elements which are larger than $\frac{a_1+a_2+a_4+a_5}{4} = \frac{11}{4}$, so after the operation, the array $a$ will become $a = [1, 3, 2]$.

Your task is to find the maximum number of elements Eshag can delete from the array $a$ by applying the operation described above some number (maybe, zero) times.

A sequence $b$ is a subsequence of an array $c$ if $b$ can be obtained from $c$ by deletion of several (possibly, zero or all) elements. The solution will be evaluated in a code environment.

---

**Training problem 3**

There are $n$ squares drawn from left to right on the floor. The $i$-th square has three integers $p_i, a_i, b_i$, written on it. The sequence $p_1, p_2,$
$dots, p_n$ forms a permutation.

Each round you will start from the leftmost square $1$ and jump to the right. If you are now on the $i$-th square, you can do one of the following two operations:

1. Jump to the $i + 1$-th square and pay the cost $a_i$. If $i = n$, then you can end the round and pay the cost $a_i$.
2. Jump to the $j$-th square and pay the cost $b_i$, where $j$ is the leftmost square that satisfies $j > i, p_j > p_i$. If there is no such $j$ then you can end the round and pay the cost $b_i$.

There are $q$ rounds in the game. To make the game more difficult, you need to maintain a square set $S$ (initially it is empty). You must pass through these squares during the round (other squares can also be passed through). The square set $S$ for . . .

---

**Training problem 4**

YouKn0wWho has an integer sequence $a_1, a_2, \ldots a_n$. Now he will split the sequence $a$ into one or more consecutive subarrays so that each element of $a$ belongs to exactly one subarray. Let $k$ be the number of resulting subarrays, and $h_1, h_2, \ldots, h_k$ be the lengths of the longest increasing subsequences of corresponding subarrays.

For example, if we split $[2, 5, 3, 1, 4, 3, 2, 2, 5, 1]$ into $[2, 5, 3, 1, 4]$, $[3, 2, 2, 5]$, $[1]$, then $h = [3, 2, 1]$.

YouKn0wWho wonders if it is possible to split the sequence $a$ in such a way that the bitwise XOR of $h_1, h_2, \ldots, h_k$ is equal to $0$. You have to tell whether it is possible.

The longest increasing subsequence (LIS) of a sequence $b_1, b_2, \ldots, b_m$ is the longest sequence of valid indices $i_1, i_2, \ldots, i_k$ such that $i_1, i_2, \ldots, i_k$ and $b_{i_1}, b_{i_2}, \ldots, b_{i_k}$. For ex …

**Training problem 5**

Eve is a beginner stand-up comedian. Her first show gathered a grand total of two spectators: Alice and Bob.

Eve prepared $a_1 + a_2 + a_3 + a_4$ jokes to tell, grouped by their type:

type 1: both Alice and Bob like them;

type 2: Alice likes them, but Bob doesn't;

type 3: Bob likes them, but Alice doesn't;

type 4: neither Alice nor Bob likes them.

Initially, both spectators have their mood equal to $0$. When a spectator hears a joke he/she likes, his/her mood increases by $1$. When a spectator hears a joke he/she doesn't like, his/her mood decreases by $1$. If the mood of a spectator becomes negative (strictly below zero), he/she leaves.

When someone leaves, Eve gets sad and ends the show. If no one leaves, and Eve is out of jokes, she also ends the show.

Thus, Eve wants to arrange her jokes in such a way that the show lasts as long as possible. Help her to calculate the maximum number of jokes she can tell before the show ends. The solution will be evalu …

**Training problem 6**

Solve the following coding problem using the programming language python:

zscoder has a deck of $n + m$ custom-made cards, which consists of $n$ cards labelled from $1$ to $n$ and $m$ jokers. Since zscoder is lonely, he wants to play a game with himself using those cards.

Initially, the deck is shuffled uniformly randomly and placed on the table. zscoder has a set $S$ which is initially empty.

Every second, zscoder draws the top card from the deck. If the card has a number $x$ written on it, zscoder removes the card and adds $x$ to the set $S$. If the card drawn is a joker, zscoder places all the cards back into the deck and reshuffles (uniformly randomly) the $n + m$ cards to form a new deck (hence the new deck now contains all cards from $1$ to $n$ and the $m$ jokers). Then, if $S$ currently contains all the elements from $1$ to $n$, the game ends. Shuffling the deck doesn't take time at all.

What is the expected number of seconds before the game ends? We can sho . . .

### Training problem 7

n pupils, who love to read books, study at school. It is known that each student has exactly one best friend, and each pupil is the best friend of exactly one other pupil. Each of the pupils has exactly one interesting book.

The pupils decided to share books with each other. Every day, all pupils give their own books to their best friends. Thus, every day each of the pupils has exactly one book.

Your task is to use the list of the best friends and determine the exchange of books among pupils after k days. For simplicity, all students are numbered from 1 to n in all tests. The solution will be evaluated in a code environment.

### Training problem 8

You are given a rooted tree, consisting of $n$ vertices. The vertices are numbered from 1 to $n$, the root is the vertex 1.

You can perform the following operation at most $k$ times:

choose an edge $(v, u)$ of the tree such that $v$ is a parent of $u$;

remove the edge $(v, u)$;

add an edge $(1, u)$ (i. e. make $u$ with its subtree a child of the root).

The height of a tree is the maximum depth of its vertices, and the depth of a vertex is the number of edges on the path from the root to it. For example, the depth of vertex 1 is 0, since it's the root, and the depth of all its children is 1.

What's the smallest height of the tree that can be achieved? The solution will be evaluated in a code environment.

### Training problem 9

Back in time, the seven-year-old Nora used to play lots of games with her creation ROBO-Head-02, both to have fun and enhance his abilities.

One day, Noras adoptive father, Phoenix Wyle, brought Nora n boxes of toys. Before unpacking, Nora decided to make a fun game for ROBO.

She labelled all n boxes with n distinct integers $a_1, a_2, \ldots, a_n$ and asked ROBO to do the following action several (possibly zero) times:

Pick three distinct indices i, j and k, such that $a_i|a_j$ and $a_i|a_k$. In other words, $a_i$ divides both $a_j$ and $a_k$, that is $a_j \bmod a_i = 0, a_k \bmod a_i = 0$.
After choosing, Nora will give the k-th box to ROBO, and he will place it on top of the box pile at his side. Initially, the pile is empty.
After doing so, the box k becomes unavailable for any further actions. Being . . .

### Training problem 10

This is an interactive problem

You are given a grid n× n, where n is odd. Rows are enumerated from 1 to n from up to down, columns are enumerated from 1 to n from left to right. Cell, standing on the intersection of row

x and column y, is denoted by (x, y).

Every cell contains 0 or 1. It is known that the top-left cell contains 1, and the bottom-right cell contains 0.

We want to know numbers in all cells of the grid. To do so we can ask the following questions:

$x_1 y_1 x_2 y_2$; where $1 \leq x_1 \leq x_2 \leq n, 1 \leq y_1 \leq y_2 \leq n$, and $x_1 + y_1 + 2 \leq x_2 + y_2$. In other words, we output two different cells $(x_1, y_1)$, $(x_2, y_2)$ of the grid such that we can get from the first to the second by moving only to the right and down, and they aren't adjacent.

As a response to such question you will be told if there exists a path between $(x_1, y_1)$ and $(x_2, y_2)$, going only to the right or down, numbers in cells of which form a palindrome.

For example, paths, shown in gr . . .

