# OpenReview forum: "Learning on the Job: Test-Time Curricula for Targeted Reinforcement Learning"
_ICLR.cc/2026/Conference — Submitted to ICLR 2026_

### Official Review · Reviewer_GZqh · 2025-10-18

**Soundness:** 3
**Presentation:** 2
**Contribution:** 2
**Rating:** 4
**Confidence:** 3

**Summary:**

The paper proposes a way to build task-specific curricula to improve fine-tuning of large models to specific tasks. The approach called TTC-RL is a relatively simple "reorganization" of tasks to be executed during the fine-tuning of the model, heavily based on a previously published approach (SIFT) to make the selection of the tasks.

**Strengths:**

- Approach is relatively simple to implement and widely applicable to pretty much any fine-tuning task
- Extensive experimental evaluation is presented showing that the approach is better than not using curriculum.

**Weaknesses:**

- There is a gap in the literature covered by the paper, especially in regards to the most related works on Curriculum for RL. The authors should start with the two main surveys in the area, as they mention in passage the original Curriculum paper for supervised learning, and then go straight to the LLM-specific methods:

Da Silva, Felipe Leno, and Anna Helena Reali Costa. "A survey on transfer learning for multiagent reinforcement learning systems." Journal of Artificial Intelligence Research 64 (2019): 645-703.

Narvekar, Sanmit, et al. "Curriculum learning for reinforcement learning domains: A framework and survey." Journal of Machine Learning Research 21.181 (2020): 1-50.

In special, there are quite a number of Curriculum approaches that explore "transferability" metrics similar to what you are trying to accomplish with SIFT, from the probably most related "transfer potential":

Silva, Felipe Leno Da, and Anna Helena Reali Costa. "Object-oriented curriculum generation for reinforcement learning." Proceedings of the 17th international conference on autonomous agents and multiagent systems. 2018.

to the more classical but still likely adaptable to modern research "learned transferability"

Sinapov, Jivko, et al. "Learning inter-task transferability in the absence of target task samples." Proceedings of the 2015 international conference on autonomous agents and multiagent systems. 2015.

- Related to the last point, the experimental evaluation shows only the proposed approach X not using curriculum, which at this point is no surprise that results in a better performance. The experimental evaluation should consider other classical autonomous curriculum definition methods to show it even makes sense to develop an approach specific for LLM to bridge the gap from simple adaptation of the older methods.

- The final main weakness of the paper is that a very small portion of the paper is dedicated to explain the actual proposed method. The method description (one of the most critical portions of the paper), is described exactly between lines 158-179, barely 20 lines!!!, many of those simply pointing to other papers on the literature (specifically GRPO and SIFT) instead of simply listing step by step all operations - I appreciate the openness in showing specifically the parts of the method inherited from the literature, however it would make it much easier for the reader to have Algorithm 1 listing all steps and the exact equations being computed during the method (especially important for GRPO because you omit the KL term and the comment related to this in line 177 can be easily missed or forgotten by the reader).

**Questions:**

No specific question

---

> ### Author Response · Authors · 2025-11-16
>
> Thank you for your review and detailed comments! We are happy to address each of the points individually.
>
> > Approach is relatively simple to implement and widely applicable to pretty much any fine-tuning task. Extensive experimental evaluation shows the advantage of using curricula.
>
> Thank you!
>
> > A very small portion is dedicated to explain the proposed method.
>
> With this work we do not introduce a new method for curricula, indeed SIFT was proposed by [6]. Our key contribution is the analysis of the novel setting and its extensive experimental evaluation in the context of reasoning tasks. To our knowledge, we are the first to show that test-time training via RL can significantly improve LLMs on reasoning tasks and we find that the previously studied TTT via SFT is ill-suited for this (Appendix A).
>
> Based on your and other reviewer’s feedback, we reorganized Section 3 of the paper to better reflect our contribution. We provide more details regarding the novelty of this work in the general response above. Thank you for this feedback!
>
> > There is a gap in the literature covered by the paper, especially in regards to the most related works on Curriculum for RL.
>
> Thank you for pointing out prior work on curricula for RL! TTC-RL builds upon the foundation of the proposed curricula in these methods, which identify relevant problems for solving the target task (e.g. [2,3]). Additionally, the automatic balancing of task difficulty proposed in Appendix C, builds on prior ideas such as task decomposition into simpler tasks (e.g. [1,2]). We appreciate the reference to foundational work in curricula for RL and surveys [4,5] and included citations to these works in our related work section.
>
> > The experimental evaluation should consider other autonomous curriculum methods.
>
> Thank you for highlighting this! Our primary contribution in this work is to introduce the novel framework of test-time training via RL for language reasoning tasks, rather than a new auto-curriculum algorithm. Hence, we aimed our experiments at showcasing the general advantages of targeted curricula instead of a particular method.
>
> Moreover, to the best of our knowledge, ours is the first work to study targeted curricula at test time for LLM reasoning, and we are not aware of existing auto-curriculum methods that can be directly adapted to this setting. In particular approaches relying on a transfer potential (e.g. [1,2]) or learned transferability functions (e.g. [3]), rely on task specific features, objects or Q-values, all of which are not available in natural language reasoning tasks. If you are aware of any approaches, which allow for adaptation to our setting, we would be happy to incorporate them into our experiments! Our evaluated curricula can be intuitively seen as descendants of methods mentioned in [1,2,3] which use the LLMs representation space to identify “similar” tasks and are added to our related work.
>
> Further, we would like to highlight Appendix C where we ablate one extension to the particular curriculum method which adaptively poses training tasks of appropriate difficulty. We did not include this in the main body to focus on an evaluation of the simplest implementation that highlights the potential of the TTC-RL setting. We find that the adaptive curriculum from Appendix C can further accelerate learning when the tasks in the corpus tend to be too difficult for the initial model (in our experiments Qwen3-0.6B).
>
> ---
>
> Thank you for your extensive feedback and mentions of prior work which has already been instrumental in improving the paper! We would be happy to hear whether our response and updates to the paper have adeptly addressed your remaining concerns and questions. Please let us know if you have any further questions!
>
> [1] Svetlik et al. Automatic Curriculum Graph Generation for Reinforcement Learning Agents. AAAI 2017.
>
> [2] Silva & Costa. Object-oriented curriculum generation for reinforcement learning. AAMAS 2018.
>
> [3] Sinapov et al. Learning inter-task transferability in the absence of target task samples. AAMAS 2015.
>
> [4] Narvekar et al. Curriculum learning for reinforcement learning domains: A framework and survey. JMLR 2020.
>
> [5] Da Silva & Costa. A survey on transfer learning for multiagent reinforcement learning systems. JAIR 2019.
>
> [6] Hubotter et al. Efficiently Learning at Test-Time: Active Fine-Tuning of LLMs. ICLR 2025.

---

> ### Author Response · Authors · 2025-11-21
>
> Dear Reviewer,
> We greatly appreciate your valuable feedback. We would like to double-check if our response was able to address your points. We also included an extensive discussion of novelty and a comparison of our results to frontier models in the general response above. Please let us know if anything remains unclear. We are happy to further discuss and add additional details to the paper!

---

### Official Review · Reviewer_JVHq · 2025-10-31

**Soundness:** 3
**Presentation:** 3
**Contribution:** 3
**Rating:** 4
**Confidence:** 3

**Summary:**

This paper introduces Test-Time Curriculum Reinforcement Learning (TTC-RL), a framework that enables large language models (LLMs) to autonomously curate task-specific curricula during test-time and improve via reinforcement learning. The core idea is to leverage a diverse corpus of training tasks, selectively retrieving those most relevant to a target task using the SIFT algorithm, and fine-tuning the model via GRPO. Experiments across math (AIME, MATH500), coding (Codeforces, CodeElo), and scientific reasoning (GPQA-D) benchmarks demonstrate significant performance gains—e.g., improving Qwen3-8B’s pass@1 on AIME25 by ~1.8× and on Codeforces by ~2.4×. The method also raises performance ceilings (e.g., pass@64 on Codeforces from 45% to 72%) and outperforms models with extended in-context reasoning. The authors argue that TTC-RL offers a compute-efficient alternative to scaling context windows.

**Strengths:**

1. The idea of test-time curriculum learning bridges the gap between large-scale pre-training and fixed test-time inference. It extends the "test-time scaling" paradigm by enabling models to meta-learn from targeted experiences, which is underexplored in LLM literature.

2. The paper provides extensive evaluations across multiple models (Qwen, Llama) and domains, showing consistent improvements over baselines like general-purpose RL post-training. The gains on competition-level benchmarks (e.g., Codeforces) are particularly impressive.

3. TTC-RL avoids human-curated data and reduces reliance on expanding context windows, highlighting a compute-optimal Pareto frontier between model specialization and resource costs.

4. The inclusion of "latent improvement" metrics addresses concerns about format overfitting, and ablations (e.g., SIFT hyperparameters, curriculum sizing) validate design choices.

**Weaknesses:**

1. Computational Cost: Training on dynamically selected curricula during test-time may incur non-trivial overhead, which is not quantified in terms of wall-clock time or energy usage.

2. Generalization Concerns: Specialization to target tasks comes at the cost of degraded performance on unrelated tasks (Figure 4), raising questions about balanced multi-task adaptation.

**Questions:**

1. You show TTC-RL beats 30k-token "thinking" models, but could in-context learning with smarter retrieval (e.g., RAG) close this gap?

2. For deployment, how would TTC-RL handle dynamically evolving tasks (e.g., user-specific queries)? Is there a risk of overfitting to transient test distributions?

---

> ### Author Response · Authors · 2025-11-16
>
> Thank you for your review and detailed comments! We are happy to address each of the points individually.
>
> > The idea of test-time curriculum learning bridges the gap between large-scale pre-training and fixed test-time inference. It extends the "test-time scaling" paradigm by enabling models to meta-learn from targeted experiences, which is underexplored in LLM literature.
>
> Thank you for highlighting the novelty of our contribution!
>
> > Computational Cost: Training on dynamically selected curricula during test-time may incur non-trivial overhead, which is not quantified in terms of wall-clock time or energy usage.
>
> We agree that RL-based adaptation can be time and resource intensive. Therefore, we apply it only to the most challenging reasoning domains, aiming to advance the capabilities of frontier open-weight models. Improving the efficiency for example through adaptive allocation of test-time compute is an exciting avenue for future work.
> All experiments were run on an HPC node with four Nvidia GH-200 GPUs. We added details regarding infrastructure and wall-clock times in Appendix E.4.
>
> Furthermore, our studied setting also extends to cases where a model is trained for a particular set of target tasks (e.g., a use case). In this case training does not happen during deployment and costs amortize. TTCs can automate the costly process of human data curation for individual use cases.
>
> > Generalization Concerns: Specialization to target tasks comes at the cost of degraded performance on unrelated tasks (Figure 4), raising questions about balanced multi-task adaptation. For deployment, how would TTC-RL handle dynamically evolving tasks (e.g., user-specific queries)? Is there a risk of overfitting to transient test distributions?
>
> Your observation that test-time training specializes the model to the specific tasks encountered during adaptation, which can lead to reduced performance on unrelated tasks is spot on! Since the test-time trained model will only be evaluated on the target task, this is typically not a concern in test-time training. The goal of test-time training is to adapt the model to a particular test task and maximize performance on that task. Furthermore, decreasing performance on other tasks can even be seen as a positive symptom of the “model reallocating capacity to concepts relevant to the test task” [1].
>
> To avoid negative transfer across tasks, our analyzed TTC-RL always starts from the initial model for each new set of target tasks. Leveraging experience accumulated during test-time training across diverse and transient test distributions for continual improvement is an open and very interesting direction for further research. We believe that this work can provide a solid foundation and experimental setting for such a study. In particular, we commit to releasing model checkpoints (e.g., for the 30 individual models trained on individual AIME25 tasks) which could be a particularly useful starting point for studying continual improvement and cross generalization.
>
> > Could in-context learning with retrieval close this gap?
>
> Good question! In-context learning in conjunction with retrieval has been shown to yield strong performance in domains, where factual knowledge is retrieved. The reasoning tasks we consider only provide a verifier if a solution is correct and therefore lack ground-truth answer traces, which could be retrieved. In our setting the TTC-RL agent practices on similar tasks with each attempt up to 8k tokens. Retrieving only 125 attempts would result in retrieving up to 1M tokens which is infeasible to put in context (the KV cache alone would have size around 150GB for Qwen3-8B). In contrast to in-context learning, test-time training enables continued learning across extensive task-related experience by sequentially compressing experience into model weights.
>
> ---
>
> Thank you again for your detailed feedback. We would be happy to hear whether our response has adeptly addressed your remaining concerns and questions. Please let us know if you have any further questions.
>
> [1]: Hubotter et al. Specialization after Generalization: Towards Understanding Test-Time Training in Foundation Models.

---

> ### Author Response · Authors · 2025-11-21
>
> Dear Reviewer,
> We greatly appreciate your valuable feedback. We would like to double-check if our response was able to address your points. Please let us know if anything remains unclear. We are happy to further discuss and add additional details to the paper!

---

### Official Review · Reviewer_3yLh · 2025-10-31

**Soundness:** 3
**Presentation:** 2
**Contribution:** 2
**Rating:** 4
**Confidence:** 4

**Summary:**

This paper proposes to use an auto-curriculum to sequence tasks for RL post training on a corpus, so that performance is improved on a set of target tasks. In particular, the paper proposes to use a previous method, SIFT, alongside a corpus of verifiable tasks that they have curated. Results across math and coding show that sequencing tasks in a curriculum leads to large improvements in tasks compared to randomly sampling these tasks. These results are contextualized by looking at performance differences across tasks, and with smaller models.

**Strengths:**

- The empirical results suggest that curricula for post-training can lead to large improvements in performance.
- Most of the paper is well-written, but some areas remain unclear (see below)

**Weaknesses:**

- Lack of novelty: performance can be attributed to a previously proposed method that is applied to generating curricula on the verifiable-corpus.
- There is comparatively little discussion of the proposed method. I also found Section 5 much less clear than the rest of the paper.
- No other auto-curriculum baseline: despite comparing the proposed curriculum method with post-training + random sampling, no other methods for constructing curricula are compared.

**Questions:**

- Figure 1: "While avoiding human-driven data curation" is not entirely a fair comparison, given that the curricula involves a corpus of data curated by humans
- Section 3.1: The main method proposed here is barely explained, and seems to be a straightforward application of existing methods.
   From the appendix, it seems
- Section 3.1: From the appendix, it seems like SIFT is similar to a leave-one out estimator. Wouldnt this be extremely costly to compute with an LLM?
- Section 3.2: by introducing a corpus along with the method, there is some possibility that this corpus is particularly suited to your method or to curricula in particular. For example, it may be the case that the curriculum is effective because uniform random sampling (as in your post-training baseline) often samples a completely irrelevant task. However, this difference would disappear if both rl post training and the curriculum is allowed to learn from the entire corpus. Of course, this would be computationally challenging to investigate.
- line 313: Can you clarify what you mean by "improvement to the RL training algorithm"? Would these improvements also be used in the rl post-training baseline? If so, then I think this discredits the referenced claim that RL training is simply distilling pass@k, and doesn't fully explain the improvements brought by the curriculum.
- line 348: Is the cost of TTC-RL linear? Isnt there a cost associated with using SIFT?
- Section 4.2: While it is good to report this finding, I think this provides some evidence that there could be potential overfitting occurring by sequencing a narrow set of tasks. One thing that doesnt seem reported is whether the model actually degrades on tasks that are very different from the target task. (Figure 4, right seems to report "normalized accuracy" but not "accuracy change")
- line 399: I don't understand the lowerbound, P(correct) > P(accurate). Wouldnt correctness require accuracy as a pre-condition? This would imply that P(accurte) > P(correct). Also, what is the difference between accurate and well-formed: they are different in eq 1 but talked about as if theyre the same on line 398.

---

> ### Author Response · Authors · 2025-11-16
>
> Thank you for your review and detailed comments! We are happy to address each of the points individually.
>
> > The empirical results suggest that TTCs can lead to large improvements in performance
>
> Thank you for highlighting the strong empirical results!
>
> > Lack of novel method & Little discussion of the proposed methods (Section 3.1)
>
> We fully agree with you that our work relies on previously introduced methodology. We view the analysis of the novel setting and its extensive experimental evaluation in the context of reasoning tasks as our key contribution. To our knowledge, we are the first to show that test-time training via RL can significantly improve LLMs on reasoning tasks and we find that the previously studied TTT via SFT is ill-suited for this (Appendix A).
>
> Based on your and other reviewer’s feedback, we reorganized Section 3 of the paper to better reflect our contribution. We provide more details regarding the novelty of this work in the general response above. Thank you for this feedback!
>
> > No other auto-curriculum baseline.
>
> Thank you for highlighting this! Our primary contribution in this work is to introduce the novel framework of test-time training via RL for reasoning tasks. To the best of our knowledge, ours is the first work to study targeted curricula at test time for LLM reasoning, and we are not aware of existing auto-curriculum methods that can be directly adapted to this setting. If you are aware of such prior work, we would be happy to incorporate it into our experiments!
>
> Further, we would like to highlight Appendix C where we ablate one extension to the particular curriculum method which adaptively poses training tasks of appropriate difficulty. We did not include this in the main body to focus on an evaluation of the simplest implementation that highlights the potential of the TTC-RL setting. We find that the adaptive curriculum from Appendix C can further accelerate learning when the tasks in the corpus tend to be too difficult for the initial model (in our experiments Qwen3-0.6B).
>
> > Figure 1: "While avoiding human-driven data curation" is not entirely a fair comparison, given that the curricula involves a corpus of data curated by humans
>
> You are completely correct. We removed this inaccurate statement from the caption. Thank you for raising this! Still, compared to repeated curation per test-time task, the initial curation of a corpus is relatively cheap. TTC-RL enables “true” test-time training where for a given “prompt” the agent autonomously sets its own practice tasks, without waiting for human intervention.
>
> > From the appendix, it seems like SIFT is similar to a leave-one out estimator. Wouldn’t this be extremely costly to compute with an LLM?
>
> We adopt the open-source implementation of SIFT from [1]. As shown in [1, Figure 4], this implementation of SIFT is only marginally slower than nearest neighbor retrieval which has been scaled to corpora of billions of documents.
>
> > Section 3.2: by introducing a corpus along with the method, there is some possibility that this corpus is particularly suited to your method or to curricula in particular. For example, it may be the case that the curriculum is effective because uniform random sampling (as in your post-training baseline) often samples a completely irrelevant task. However, this difference would disappear if both rl post training and the curriculum is allowed to learn from the entire corpus. Of course, this would be computationally challenging to investigate.
>
> We did not tune the corpus to match our setting. Our corpus is merely a combination of existing corpora that were created for math / coding / general reasoning, with a few additional quality filters (cf. Appendix E.1). We agree that it would be interesting to train a model on the full corpus as a baseline, but this would exceed our computational budget. We expect, however, that tasks from our verifiable-corpus are also well-represented in post-training pipelines that were used in the training of our tested initial models.
>
> There is recent work that analyzes test-time training in comparison to training on a full dataset, in a computationally lighter setting [2].
>
> Perhaps partially addressing your concern, we included an ablation in Figure 10 of the appendix where we compare TTC-RL, RL post-training and RL post-training restricted to the code subset of our corpus.

---

> > ### Author Response · Authors · 2025-11-16
> >
> > > line 313: Can you clarify what you mean by "improvement to the RL training algorithm"? Would these improvements also be used in the rl post-training baseline? If so, then I think this discredits the referenced claim that RL training is simply distilling pass@k, and doesn't fully explain the improvements brought by the curriculum.
> >
> > Thank you for this question! Our RL post-training baseline uses exactly the same parameters as TTC-RL and only differs in the training tasks. The mentioned “improvements” refer to increasing in the clip-high parameter in the GRPO algorithm which leads to improved exploration and prevents entropy collapse (cf. Appendix D.1). In Appendix D.1, we even show that tuning clip-high can further improve upon the results reported in the main paper, improving Qwen3-8B on Codeforces to 35%.
> >
> > We agree with your assessment that with an increased clip-high neither RL post-training nor TTC-RL are simply distilling pass@k. While the improvement of TTC-RL over RL post-training is entirely due to the selected data, the adjustment to clip-high is also crucial for its gains in pass@k.
> >
> > We rephrased the main text to be more specific, thank you for raising this!
> >
> > > line 348: Is the cost of TTC-RL linear? Isn’t there a cost associated with using SIFT?
> >
> > This is a really great question! Our data selection implementation works as follows: Before training, we first pre-select tasks from the corpus via nearest neighbor retrieval. We then use the implementation of SIFT described in Appendix H.1 of [1] to sequentially select from the pre-selected tasks. As described in [1], this is linear in the number of selected tasks. We added a paragraph to Section B.1 describing the computational complexity of SIFT.
> >
> > > Section 4.2: While it is good to report this finding, I think this provides some evidence that there could be potential overfitting occurring by sequencing a narrow set of tasks. One thing that doesn’t seem reported is whether the model actually degrades on tasks that are very different from the target task.
> >
> > This is a great observation! Since the test-time trained model will only be evaluated on the target task, this is typically not a concern in test-time training. Furthermore, decreasing performance on other tasks can even be a positive symptom as the “model reallocates capacity to concepts relevant to the test task” [2]. When given the next test task, test-time training will restart from the initial model to prevent the “forgetting” you are rightly concerned of.
> >
> > In our view, leveraging experience accumulated during test-time training across diverse and transient test distributions for continual improvement is an open and very interesting direction for further research. We believe that this work can provide a solid foundation and experimental setting for such a study.
> >
> > > I don't understand the lowerbound, $P(\text{correct}) > P(\text{accurate})$.
> >
> > We define the event “accurate” as the intersection of the events “correct” and “well-formed”. Hence, $P(\text{accurate}) = P(\text{correct} \cap \text{well-formed}) < P(\text{correct})$ since $\text{accurate} \subseteq \text{correct}$. Intuitively, the probability of an answer being correct is larger than it being marked as accurate, as it might not be well-formated but still correct. Conversely, if an answer is marked as accurate then it is guaranteed to be correct. We hope this clarifies the lower bound.
> >
> > > I also found Section 5 much less clear than the rest of the paper.
> >
> > We would be happy to clarify any remaining questions and improve Section 5!
> >
> > ---
> >
> > Thank you for your extensive feedback which has already been instrumental in improving the paper! We would be happy to hear whether our response and updates to the paper have adeptly addressed your remaining concerns and questions. Please let us know if you have any further questions!
> >
> > [1]: Hubotter et al. Efficiently Learning at Test-Time: Active Fine-Tuning of LLMs. ICLR 2025.
> >
> > [2]: Hubotter et al. Specialization after Generalization: Towards Understanding Test-Time Training in Foundation Models.

---

> ### Author Response · Authors · 2025-11-21
>
> Dear Reviewer,
> We greatly appreciate your valuable feedback. We would like to double-check if our response was able to address your points. We also included an extensive discussion of novelty and a comparison of our results to frontier models in the general response above. Please let us know if anything remains unclear. We are happy to further discuss and add additional details to the paper!

---

### Official Review · Reviewer_mSgC · 2025-10-31

**Soundness:** 3
**Presentation:** 3
**Contribution:** 2
**Rating:** 6
**Confidence:** 3

**Summary:**

This paper proposes Test-Time Curriculum Reinforcement Learning (TTC-RL), a method for specializing large language models (LLMs) to target tasks at test time. The model automatically constructs a self-curated curriculum from a large training corpus (via SIFT embeddings) and fine-tunes itself using on-policy RL (GRPO).

**Strengths:**

- The idea of the paper, automatic self-curated test-time curricula (TTC-RL) for LLM specialization via RL, is novel.
- Strong empirical gains: Significant improvements across math and reasoning benchmarks done sample-efficiently.
- The Latent Improvement (LI) to measure real reasoning gains, not just formatting is intuitive and nicely thought.
- Reproducibility: Public verifiable-corpus (~280k tasks) and clear training pipeline.

**Weaknesses:**

- Primarily focuses on non-thinking models (Qwen3-8B, Llama-3.1-8B).
- TTC-RL relies on a fixed training corpus, limiting potential performance on completely novel target tasks.
- While more efficient than long-context scaling, RL-based test-time specialization still requires on-the-fly training, which could be impractical in some real-world applications.
- No code release: Makes it harder for the community to reproduce or extend the TTC-RL pipeline.

**Questions:**

- How does TTC-RL perform when target tasks are very different from training corpus tasks (e.g., out-of-distribution problems)?
- Can TTC-RL combine multiple target tasks with diverse difficulty levels effectively, or does it favor easier tasks?

---

> ### Author Response · Authors · 2025-11-16
>
> Thank you for your review, and the positive evaluation of our work. We are happy to address each point.
>
> > Automatic self-curated TTC-RL for LLM specialization via RL is novel. Strong empirical gains: Significant improvements across math and reasoning benchmarks done sample-efficiently.
>
> Thank you!
>
> > The Latent Improvement (LI) to measure real reasoning gains, not just formatting is intuitive and nicely thought.
>
> Thank you for highlighting the contribution of the LI measure!
>
> > Primarily focuses on non-thinking models.
>
> We agree that the evaluation of TTC-RL on thinking models would be particularly interesting and would further add to the empirical validation of the setting. Unfortunately, training thinking models requires substantial compute & GPU memory due to substantially longer generations. This makes an evaluation of TTC with thinking models infeasible for us. However, since thinking models differ from non-thinking models primarily in their generation length, we would expect our results to translate to thinking models. Further, we compare to thinking models in Figure 3 and show that TTC teaches the model to use its smaller context more effectively, *remarkably, closing the gap to the thinking model* in coding and GPQA.
>
> > TTC-RL relies on a fixed training corpus / How does TTC-RL perform on very different tasks?
>
> Thank you for raising this important point! TTCs indeed rely heavily on the utility of the training corpus for the target tasks, which prevents the immediate application to completely novel target tasks. This limitation reveals exciting avenues for future work, where training tasks may be generated instead of selected. We believe that our results and artifacts (code, data, models) can serve as a foundation for such future work.
>
> > RL-based test-time scaling requires on-the-fly training, which could be impractical.
>
> We agree that RL-based adaptation can be time and resource intensive. Therefore, we apply it only to the most challenging reasoning domains, aiming to advance the capabilities of frontier open-weight models. Improving the efficiency for example through adaptive allocation of test-time compute is an exciting avenue for future work.
>
> > Can TTC-RL combine target tasks of varying difficulty level?
>
> Our target benchmark datasets (e.g. MATH500, CodeELO) combine questions of a wide range of difficulties. While naive nearest-neighbor selection favours selecting tasks, which are close in the embedding space to any of the target tasks, we utilize SIFT. SIFT not only optimizes for proximity but also for diversity in the selected tasks, naturally selecting useful training tasks for target tasks of varying difficulty.
>
> > No code release
>
> The full TTC-RL codebase and scripts are provided in the supplementary materials for full reproducibility. In addition we are committed to release models, experimental logs and the verifiable-corpus to facilitate future research.
>
> We would be happy to hear whether our response has adeptly addressed your remaining concerns and questions. Please let us know if you have any further questions!

---

> > ### Author Response · Authors · 2025-11-21
> >
> > Dear Reviewer,
> > We greatly appreciate your valuable feedback. We would like to double-check if our response was able to address your points. Please let us know if anything remains unclear. We are happy to further discuss and add additional details to the paper!

---

> > > ### Comment · Reviewer_mSgC · 2025-11-23
> > >
> > > Thank you for the clarifications. The explanations are satisfactory, and I remain positive about the work. I will keep the score I originally gave.

---

> > > > ### Author Response · Authors · 2025-11-26
> > > >
> > > > Thank you for your engagement during the discussion period and for maintaining your positive score. We appreciate your feedback and remain available if you have any further questions.

---

### Official Review · Reviewer_Uw6i · 2025-10-31

**Soundness:** 3
**Presentation:** 4
**Contribution:** 2
**Rating:** 6
**Confidence:** 3

**Summary:**

This paper proposes a test-time curriculum method for RL-based fine-tuning of large language models. In contrast to manually curated datasets, TTC-RL selects relevant examples in a corpus that are closest to the test-time task based on their last-token embeddings, then fine-tunes the model using the selected samples. Experiments across a variety of benchmarks in math, coding, and scientific reasoning demonstrate that TTC-RL improves performance in pass@k, with k ranging from 1 to 64.

**Strengths:**

- The paper is very well-written, has an easy-to-follow structure, and includes figures that well describe the empirical advantages of the proposed framework.
- The proposed framework comes from an existing idea proposed for SFT, yet this work seems to be the first to apply this idea to test-time training via RL-based fine-tuning.
- Experiments cover a wide range of benchmarks, hence they provide comprehensive evidence that shows the advantages of TTC-RL in multiple base models. The results are well-documented as well.

**Weaknesses:**

- Although this is the first application of the nearest neighbor retrieval for sample selection in test-time training via RL, the idea is not novel at all.
- Although it makes sense to select samples from a corpus based on the similarity of last-token embeddings, the generated curriculum is fixed throughout test-time training. It does not depend on how the model performs at all, either.

**Questions:**

- Despite being knowledgeable about curriculum learning for RL and RLHF, I do not know much about test-time training. What is the practical purpose of such a framework, considering that at "test-time" there shouldn't be any training?
- The proposed approach selects samples based on embedding similarity, but this set is fixed during test-time training. Has it ever been considered to adaptively prioritize certain samples during TTC?

---

> ### Author Response · Authors · 2025-11-16
>
> Thank you for your review, and the positive evaluation of our work. We are happy to address each point.
>
> > The paper is very well-written, and includes figures that well describe the empirical advantages of the proposed framework. Experiments cover a wide range of benchmarks, hence they provide comprehensive evidence that shows the advantages of TTC-RL in multiple base models. The results are well-documented as well.
>
> Thank you for this positive evaluation!
>
> > Although this is the first application of NN retrieval for TTT via RL, the idea is not novel at all.
>
> We fully agree with you that our work relies heavily on previously introduced methodology. We view the analysis of the novel setting and its extensive experimental evaluation in the context of reasoning tasks as our key contribution. As you emphasize in your review, we are the first to show that test-time training via RL can significantly improve LLMs on reasoning tasks and we find that the previously studied TTT via SFT is ill-suited for this (Appendix A).
>
> Based on your and other reviewer’s feedback, we reorganized Section 3 of the paper to better reflect our contribution. We provide more details in the general response above. Thank you for this feedback!
>
> > The generated curriculum is fixed throughout TTT. Have you considered adaptively prioritizing certain samples?
>
> Thank you for highlighting this! The curriculum we evaluate primarily is indeed fixed throughout TTT. Intuitively, however, adaptively balancing the curriculum to pose tasks that are neither “too easy” nor “too hard” should lead to faster learning. We evaluate an adaptive curriculum in Appendix C, which we find to accelerate learning for a weaker model (Qwen3-0.6B) where the difficulty of the corpus is “misaligned” with respect to the model’s initial capability. We focused the main evaluation on the simpler static variant, since the adaptive variant did not improve upon a static curriculum for Qwen3-8B (and similarly strong models). We hypothesize that this is because most tasks within the corpus are already of appropriate difficulty for models of this capability.
>
> Based on your feedback, we added an explicit mention of our analysis of adaptive curricula to Section 3, highlighting the flexibility for curricula.
>
> > What is the practical purpose of test-time training, considering that at "test-time" there shouldn't be any training?
>
> Thank you for this question! Test-time training (TTT) is an emerging approach enabling models to continue learning/improving for a given prompt or group of prompts (i.e., at “test-time”). TTT addresses the memory bottleneck of self-attention in a transformer that effectively prevents scaling in-context learning to long sequences. For example, the extensive targeted interaction with an environment as in this work is not feasible to perform solely in-context, since we reach “context lengths” of several million tokens (where the KV cache size of Qwen3-8B would be well above 100GB). To avoid the memory bottleneck, experience has to be compressed into a finite state. TTT uses the model itself as the state [1,2] and updates the state/model via gradient descent. This applies the same method that is commonly used at train-time to compress data, but applies it at test-time for targeted learning towards a particular task.
>
> In this way, TTT can be thought of as a scalable way to solve challenging tasks, which a base model cannot solve within a computationally feasible context window [e.g., see also 3]
>
> Thank you again for your review, which we believe has been instrumental in improving our paper!
> We would be happy to hear whether our response and updates to the paper have adeptly addressed your remaining concerns and questions. Please let us know if you have any further questions!
>
> ---
>
> [1]: Sun et al. Learning to (Learn at Test Time): RNNs with Expressive Hidden States. ICML 2025.
>
> [2]: Hardt & Sun. Test-Time Training on Nearest Neighbors for Large Language Models. ICLR 2024.
>
> [3]: Zuo et al. TTRL: Test-Time Reinforcement Learning. NeurIPS 2025.

---

> ### Author Response · Authors · 2025-11-21
>
> Dear Reviewer,
> We greatly appreciate your valuable feedback. We would like to double-check if our response was able to address your points. We also included an extensive discussion of novelty and a comparison of our results to frontier models in the general response above. Please let us know if anything remains unclear. We are happy to further discuss and add additional details to the paper!

---

### Author Response · Authors · 2025-11-16
**General response**

Dear Reviewers,

We thank you for your detailed reviews and comments. Several reviews raised the missing methodological novelty of our work, which we would like to address here.

## Novelty of this work

Our aim with this work is **not to propose a new method for curriculum design**, but rather to experimentally analyze a novel setting. We analyze a targeted RL setting where an LLM is trained at test-time for a particular task. To the best of our knowledge this setting has not been studied before in the context of complex reasoning tasks (math, coding, general reasoning) and LLMs. Prior work has either focused on different families of tasks or used SFT instead of RL, which we find not to improve downstream task performance. To this end, this work shows the advantages of the targeted RL setting for reasoning, rather than analyzing the advantage of a particular curriculum method over another curriculum method.

Following your feedback, we extensively restructured Section 3 to emphasize the novelty of our setting and findings while clearly attributing prior methodological work and citing your mentioned prior work. We believe that this restructuring portrays our contributions more accurately and thank all reviewers for their instrumental and important feedback!

## Summary of our experimental findings

We thank all reviewers for **highlighting the experimental strengths of this paper** (which we view as our main contribution):

* This work is the first work to study test-time training via RL on training tasks similar to test-time tasks (Uw6i, mSgC, JVHq)
* Strong experimental results across diverse benchmarks with well-documented results (Uw6i, mSgC, 3yLh, JVHq, GZqh)

We are happy to see that reviewers almost unanimously point out the two core contributions of this paper.

In our view, the key contribution of this work is the systematic experimental investigation of a previously unstudied setting. We find that targeted RL with LLMs **leads to remarkable improvements in reasoning on challenging tasks** on top of **strong baselines such as (size-relative) frontier post-trained open-source models** like Qwen3-4B-2507.
For context, we emphasize that training Qwen3-8B in our proposed test-time setting reaches scores close to recent frontier models across several benchmarks. **For example, Qwen3-8B + TTC-RL approximately matches GPT-4o-2024-08-06 on [LCBv6](https://livecodebench.github.io/leaderboard.html) and outperforms GPT 4.1 and Claude Opus 4.1 on [AIME](https://www.vals.ai/benchmarks/aime).**

Our findings indicate that TTC-RL can be an effective setting for improving LLMs on reasoning tasks.
We include our complete code implementation with this submission and also commit to openly releasing the corpus, model checkpoints, and W&B training logs to foster future work in this setting.

---

> ### Author Response · Authors · 2025-11-16
>
> ## Detailed comparison to frontier models
>
> | Model                      | [LCBv6 (02/25–05/25)](https://livecodebench.github.io/leaderboard.html) | Size (B params) |
> |----------------------------|----------------------------------------------------------------------------|-----------------|
> | Claude-3.5-Sonnet-20241022 | 36.0                                                                       | —               |
> | GPT-4O-2024-08-06          | 29.5                                                                       | —               |
> | GPT-4-Turbo-2024-04-09     | 28.9                                                                       | —               |
> | **TTC-RL** (Qwen3-8B)      | 27.3                                                                       | **8**             |
> | **TTC-RL** (Qwen3-4B)      | 26.9                                                                       | **4**             |
> | GPT-4o-mini-2024-07-18     | 26.8                                                                       | —               |
> | Claude-3-Haiku             | 24.5                                                                       | —               |
> | DeepDeek V3                | 16.0                                                                       | 671 (37 active)        |
>
> | Model                    | [MATH500](https://www.vals.ai/benchmarks/math500-08-26-2025) | Size (B params) |
> |--------------------------|--------------------------------------------------------------|-----------------|
> | Grok 3                   | 89.8                                                        | —            |
> | DeepSeek V3 (03/24/2025) | 88.6                                                        | 671 (37 active)        |
> | **TTC-RL** (Qwen3-4B)    | 88.5                                                        | **4**             |
> | GPT 4.1                  | 87.2                                                        | —               |
> | Llama 4 Maverick         | 85.2                                                        | 400 (17 active)               |
> | **TTC-RL** (Qwen3-8B)    | 85.1                                                        | **8**             |
> | Gemini 1.5 Pro           | 82.8                                                        | —               |
> | Deepseek V3              | 80.4                                                        | 671 (37 active)        |
>
> | Model                    | [AIME24/25](https://www.vals.ai/benchmarks/aime) | Size (B params) |
> |--------------------------|--------------------------------------------------|-----------------|
> | **TTC-RL** (Qwen3-4B)    | 52.92                                            | **4**             |
> | DeepSeek V3 (03/24/2025) | 52.2                                             | 671 (37 active)        |
> | **TTC-RL** (Qwen3-8B)    | 46.25                                            | **8**             |
> | Claude Opus 4.1          | 44.2                                             | —               |
> | GPT 4.1                  | 39.6                                             | —               |
> | Claude Sonnet 4          | 38.5                                             | —               |
> | Deepseek V3              | 27.5                                             | 671 (37 active)        |
> | Llama 4 Maverick         | 25.2                                             | 400 (17 active)               |
>
> ## Additional changes
>
> We made the following changes in addition to addressing your feedback regarding novelty:
>
> * We replaced figure 2 with a figure comparing performance of TTC-RL to RL post-training across the number of training tasks. We anticipate that this figure is more informative than the previous figure 2, which we moved to the appendix.
> * We identified a previously unknown flaw in the evalchemy code evaluation & Codeforces and CodeElo datasets. We describe this flaw in lines 464-468. After addressing this flaw, we reran all code experiments, which did not meaningfully change results.

---

### Author Response · Authors · 2025-11-29
**Summary of discussion period**

Dear AC,

Below we would like to succinctly summarize the discussion period.

Reviewers have unanimously emphasized the strength of our empirical contribution. In particular:
* This work studies the novel setting of test-time training via targeted RL on training tasks similar to test-time tasks (Uw6i, mSgC, JVHq).
* We provide strong experimental results across diverse benchmarks with well-documented results (Uw6i, mSgC, 3yLh, JVHq, GZqh).

This targeted RL leads to **substantial improvements of LLMs in reasoning on challenging tasks** on top of strong baselines such as Qwen3-8B or Qwen3-4B-2507. These models, when combined with targeted RL, **reach scores close to recent frontier models across diverse benchmarks**. For example, Qwen3-8B + TTC-RL approximately matches GPT-4o-2024-08-06 on LCBv6 (competitive programming) and outperforms GPT 4.1 and Claude Opus 4.1 on AIME (competition math). We provide further details in our general response to the reviews below.

The primary concern of reviewers (Uw6i, 3yLh, GZqh) centered around the novelty of our curriculum construction. We clarified that our contribution lies in exploring a novel test-time training setting while *reusing* standard curriculum selection, rather than proposing a new selection method. We have extensively revised Section 3 to accurately portray the contribution of this work, while acknowledging prior work. While this misunderstanding prompted requests for additional baselines, none of the methods mentioned by reviewers are directly applicable to our setting. We look forward to the development of novel methods for test-time curriculum design that may be developed using this work as a foundation.

Apart from this concern around presentation, which we addressed in our revision, we were happy to see that reviewers were unanimously supporting our contributions. We will open-source all code, data, model artifacts, and several training logs to support further exploration of targeted RL with LLMs.

---

### Meta-Review · Area_Chair_H5sQ · 2026-01-06

**Summary:**

In this paper, the authors proposed a method for automatic data curation. For given target tasks D* and a large corpus of training tasks D, the paper proposes an automatic selection approach for GRPO-based post-training of LLMs for D* based on SIFT, and showed their approach yields performance gains over simple RL post-training.

**Reviewer Concerns:**

Althoug the proposed method yields performance gains over basic RL training of LLMs for reasoning, the reviewers raise several concerns regarding the paper:
- Comparison baselines: There exists other data curation methods for similar setup and further comparison is required.
- Selected TTC is static for a given task, which is only supported by their finding, no anayltic reasons.
- Primarily focuses on non-thinking models (Qwen3-8B, Llama-3.1-8B)
- Most importantly, lact of novelty. Although the proposed method shows strong performance, it is  a simple application and combination of the existing method: GRPO + SIFT. Furthermore, the main core of the method, i.e., the selection metric SIFT is even from the existing work (Hubotter et al.). This significantly diminishes the novelty of this work.

**Reviewer Scores:**

Reviewer JVHq: 4, Reviewer mSgC: 6, Reviewer GZqh: 4, Reviewer 3yLh: 4,  Reviewer Uw6i: 6

Two reviewers with score 6 seem to keep their scores and it is likely that the remaining three reviewers with score 4 still keep their scores. The main reason for their negative evaluation is lack of novelty in this work, which seems difficult to be changed.

---

### Decision · Program_Chairs · 2026-01-26

Reject